# Axial contraction and short-range compaction of chromatin synergistically promote mitotic chromosome condensation

Tom Kruitwagen[1], Annina Denoth-Lippuner[1], Bryan J Wilkins[2], Heinz Neumann[2], Yves Barral[1]*

[1]Institute of Biochemistry, Department of Biology, Eidgenössische Technische Hochschule Zürich, Zürich, Switzerland; [2]Free Floater (Junior) Research Group "Applied Synthetic Biology," Institute for Microbiology and Genetics, Georg-August University Göttingen, Göttingen, Germany

*For correspondence: yves.barral@bc.biol.ethz.ch

Competing interests: The authors declare that no competing interests exist.

**Abstract** The segregation of eukaryotic chromosomes during mitosis requires their extensive folding into units of manageable size for the mitotic spindle. Here, we report on how phosphorylation at serine 10 of histone H3 (H3 S10) contributes to this process. Using a fluorescence-based assay to study local compaction of the chromatin fiber in living yeast cells, we show that chromosome condensation entails two temporally and mechanistically distinct processes. Initially, nucleosome-nucleosome interaction triggered by H3 S10 phosphorylation and deacetylation of histone H4 promote short-range compaction of chromatin during early anaphase. Independently, condensin mediates the axial contraction of chromosome arms, a process peaking later in anaphase. Whereas defects in chromatin compaction have no observable effect on axial contraction and condensin inactivation does not affect short-range chromatin compaction, inactivation of both pathways causes synergistic defects in chromosome segregation and cell viability. Furthermore, both pathways rely at least partially on the deacetylase Hst2, suggesting that this protein helps coordinating chromatin compaction and axial contraction to properly shape mitotic chromosomes.

## Introduction

The DNA molecule at the core of any eukaryotic chromosome is a hundred to million times longer than the average diameter of the cell that hosts it. Thus, cells need to fold their genetic material in order to fit it in the interphase nucleus; they need to pack it further during mitosis, in order to move sister-chromatids safely and symmetrically apart. Furthermore, chromatin folding must be dynamic to allow transcription and replication during interphase, and such that exceptionally large chromosomes can hyper-condense during anaphase in order to fit the size of the spindle and prevent chromosome missegregation (*Neurohr et al., 2011*; *Titos et al., 2014*). Moreover, mitotic condensation also facilitates the decatenation of sister chromatids during their separation (*Charbin et al., 2014*), and might help to 'cleanse' chromosomes from transcription, replication and cohesion factors (*Yanagida, 2009*). This is thought to 'reset' the transcriptional state of genes, and prevent displaced factors from interfering with chromosome segregation. However, despite their importance for chromosome segregation, the events ensuring the mitotic condensation of chromosomes are still only partially understood.

**eLife digest** DNA in humans, yeast and other eukaryotic organisms is packaged in structures called chromosomes. When a cell divides these chromosomes are copied and then the matching pairs are separated so that each daughter cell has a full set of its genome. To enable these events to take place, the DNA must become more tightly packed so that the chromosomes become rigid units with projections called arms. Any failure in this chromosome "condensation" leads to the loss of chromosomes during cell division.

Within a chromosome, sections of DNA are wrapped around groups of proteins to make a series of linked units called nucleosomes, which resemble beads on a string. These units and other scaffold proteins together make a structure called chromatin and establish the overall shape of the chromosome. However, it is not exactly clear how the nucleosomes and scaffold proteins are rearranged during condensation.

Kruitwagen et al. used microscopy to study chromosome condensation in budding yeast. The experiments reveal that condensation involves two separate processes. First, modifications to the nucleosomes result in these units becoming more tightly packed in a process called short-range compaction. Second, a group of proteins called condensin is responsible for rearranging the compacted chromatin to enforce higher-order structure on the arms of the condensed chromosome (long-range contraction). Further experiments suggest that an enzyme called Hst2 may help to co-ordinate these processes to ensure that chromosomes adopt the right shape before the cell divides. For example, Hst2 ensures that longer chromosomes condense more than shorter ones.

A future challenge will be to find out whether chromosome condensation works in a similar way in humans and other large eukaryotes, which form much larger chromosomes with more complicated structures than yeast.

Early studies made evident that nucleosomes play a critical role in DNA packaging. In favor of the idea that they play specific roles in chromatin condensation, histone H3 is phosphorylated by aurora B throughout mitosis on a serine at position 10 in most, if not all, eukaryotes. Furthermore, aurora B inactivation leads to chromosome condensation and segregation defects in budding yeast (*Lavoie et al., 2004*), fission yeast (*Petrova et al., 2013*; *Tada et al., 2011*), HeLa cells (*Tada et al., 2011*) and roundworms (*Hagstrom et al., 2002*). However, the precise role of H3 S10 phosphorylation has remained unclear (*Ajiro and Nishimoto, 1985*). Recent data demonstrated that H3 S10 phosphorylation promotes the recruitment of the sirtuin-related deacetylase Hst2, which in turn deacetylates, at least, lysine 16 of histone H4 (*Wilkins et al., 2014*). This unmasks a basic patch, allowing H4 to interact with the acidic patch on H2A, most probably on an adjacent nucleosome (*Robinson et al., 2008*; *Gordon et al., 2005*). Thus, this cascade of events initiated by H3 phosphorylation is thought to tighten the interaction between neighboring nucleosomes. However, all studies carried out so far have failed to reveal strong phenotypes for H3 serine 10 to alanine mutations in a plethora of model organisms (*de la Barre et al., 2001*; *Afonso et al., 2014*; *Ditchfield et al., 2003*). Furthermore, mutation of this residue in budding yeast did not affect axial contraction of chromosomes and the condensation of the rDNA during regular mitoses (*Neurohr et al., 2011*; *Lavoie et al., 2004*; *Lavoie et al., 2002*). Indeed, the only phenotype identified upon replacement of H3 S10 with alanine in yeast so far is limited to the reduced ability to hyper-condense artificially long chromosomes in order to fit them in the spindle (*Neurohr et al., 2011*). Thus, it remains unclear whether H3 phosphorylation and H4 deacetylation play any general role in mitotic chromosome condensation.

The discovery that mitotic extracts of frog eggs lacking any one of the subunits of a protein complex called condensin largely failed to condense chromosomes (*Hirano and Mitchison, 1994*) opened new perspectives for understanding chromosome condensation (*Piazza et al., 2013*; *Thadani et al., 2012*). Condensin is a ring-shaped pentameric protein complex. The core of the ring is formed by two structural maintenance of chromosome (SMC) subunits, Smc2 and Smc4. Three non-SMC proteins (Brn1, Ycg1 and Ycs4 in budding yeast) close the ring. The mechanism of condensin loading on chromatin is not understood, but seems to depend on the activity of the kinase aurora

B (*Lavoie et al., 2004*; *Tada et al., 2011*). Furthermore, the non-SMC subunits were recently shown to directly bind DNA (*Piazza et al., 2014*), potentially followed by topological entrapment of chromatin inside the condensin ring (*Cuylen et al., 2011*). How condensin performs its functions in chromosome condensation is unclear, but it has been proposed that condensin's role might be structural, by inducing loops within the same DNA strand (*Cuylen et al., 2011*; *Cuylen and Haering, 2011*) or might be enzymatic by promoting positive DNA supercoiling (*Baxter and Aragón, 2012*), both assisting in a decrease in length of mitotic chromatids.

Mitigating the central role of condensin in chromosome condensation, however, were observations in model organisms as diverse as fission yeast, fly, chicken and mammalian cells that indicate that chromosomes can still, at least partially, condense in the absence of condensin (*Petrova et al., 2013*; *Coelho et al., 2003*; *Vagnarelli et al., 2006*; *Gerlich et al., 2006*). Thus, although condensin was established as a key player in chromosome condensation, it cannot be the sole factor shaping mitotic chromosomes.

In order to gather insights into whether and how H2A-H4 interaction contributes to the organization of mitotic chromosomes, we sought for a method to assay the condensation state of chromatin in vivo. Here, we use a fluorescence-based assay to investigate short-range chromatin compaction and use it to study the relationships between condensin and histone modifications during chromosome condensation in mitotic cells.

## Results

### A microscopy-based assay to measure chromatin fiber compaction

In order to develop a chromatin condensation assay, we reasoned that increased nucleosome-nucleosome interaction might render chromatin less accessible to DNA-binding proteins. To test this idea directly, we asked whether chromatin condensation restricted access for heterologous reporter proteins to their binding sites when those are introduced at a chosen chromosomal locus. Therefore, we used a yeast strain in which a set of Tet operator (TetO) repeats are inserted at the *TRP1* locus on chromosome IV, 15 kb from *CEN4*, and constitutively expressing the TetR-mCherry fusion protein, which efficiently binds the TetO repeat. As a consequence, these cells exhibit a red dot in their nucleus throughout the cell cycle (*Figure 1A*). To test whether the intensity of TetR-mCherry fluorescence possibly varied over the cell cycle, we measured the fluorescence intensity of this dot in G1 cells (unbudded), when the chromatid is decondensed, but not replicated yet, and in late anaphase mother cells, when the chromatid is separated from its sister and has reached full condensation (*Figures 1B* and 3C; (*Neurohr et al., 2011*; *Sullivan et al., 2004*; *D'Amours et al., 2004*) and see below). After subtracting background fluorescence, we noticed a highly significant (p<0.0001), 2–2.5-fold decrease in mCherry fluorescence intensity at the TetO repeats on the anaphase compared with the G1-phase chromosomes (*Figure 1A, B*).

We next asked whether the variations of fluorescence intensity at the TetO repeats reflected changes in H2A/H4 interaction. Supporting this view, mutating key residues in the H3 phosphorylation and H4 deacetylation pathway established by (*Wilkins et al., 2014*) affected these variations (*Figure 1B,C*). Strikingly, mutations that abrogate the mitotic interaction between H2A and H4, such as *H3 S10A, hst2Δ* and the *H4 Δ9–16* mutations, all abolished the reduction in brightness of the TetR-mCherry focus normally observed in anaphase cells (*Figure 1B*). In reverse, mutations that promote constitutive H2A/H4 interaction, such as *H3 S10D* and *H4 K16R,* caused the TetR-mCherry focus to constitutively show, that is, even in G1 cells, the low fluorescence intensity normally specific of anaphase cells. The effect of the *H3 S10D* mutation was indeed mediated by the recruitment of Hst2, since the *hst2Δ* mutation suppressed it; the *H3 S10D hst2Δ* double mutant cells showed constitutive high brightness, similar to *hst2Δ* single mutant cells. Interestingly, however, introducing the *H4 K16R* mutation in the *hst2Δ* mutant cells did not restore the intensity drop normally observed during anaphase, suggesting that H4 K16 is not the sole residue that Hst2 deacetylates to promote nucleosome-nucleosome interaction (*Figure 1B*).

To test whether the observed fluctuations in fluorescence intensity were specific for the *TRP1* locus or TetO/TetR-mCherry, we also measured the fluorescence intensity at the *LYS4* locus, in the middle of the right arm of chromosome IV, where we integrated LacO repeats in cells expressing LacI fused to Green Fluorescent Protein (GFP). Although the effect was slightly less pronounced, we

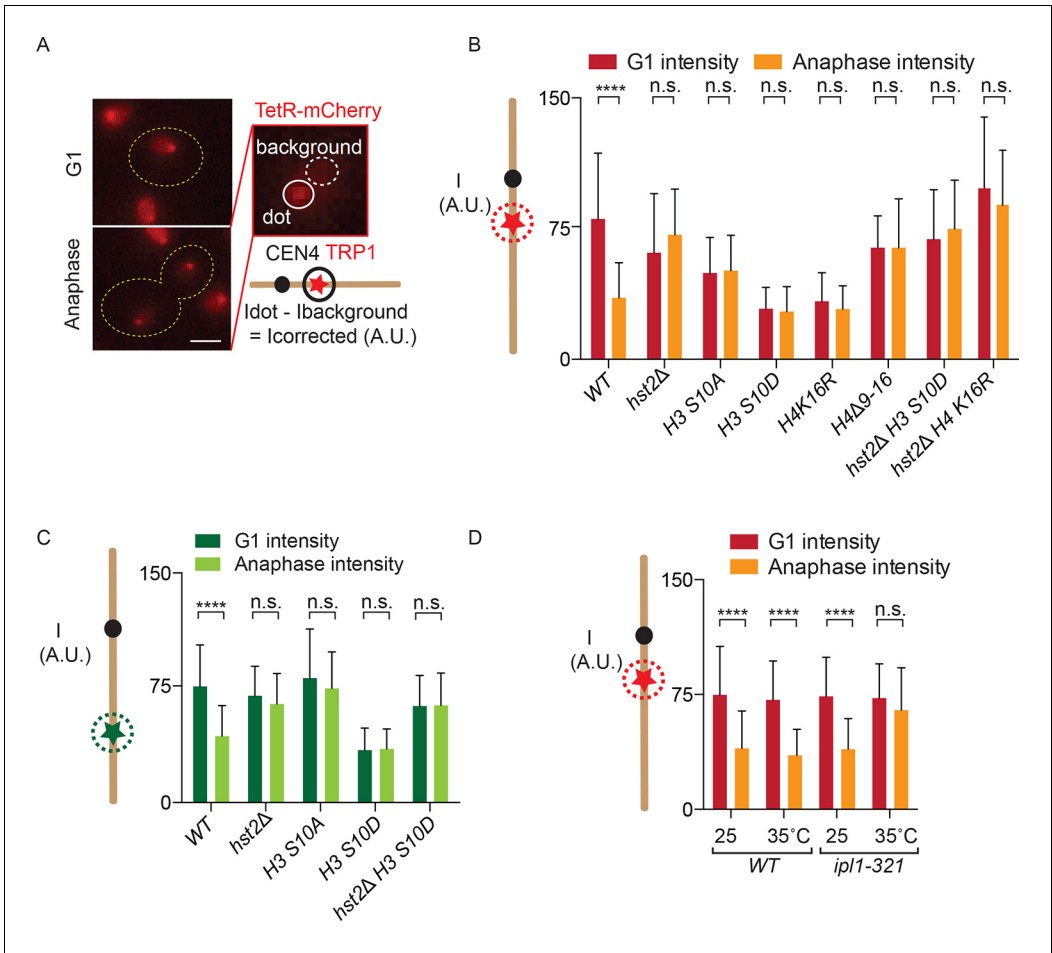

**Figure 1.** Fluorescence intensity of TetO/TetR-mCherry as a read-out for chromatin compaction. (**A**) Representative images of a cell in G1 and anaphase, containing a TetO array at the *TRP1* locus and expressing TetR-mCherry (red). Fluorescence intensity of a focus is measured by determining the total fluorescence and subtracting the background, giving the corrected fluorescence intensity. Scale bar is 2 µm. (**B**) TetR-mCherry intensities for the indicated wild type (*WT*) and mutant strains in G1 and anaphase mother cells. One way Analysis Of Variance ( ANOVA) was performed to test significance. (**C**) Fluorescence intensity for a wild type strain containing *LYS4:LacO* and expressing LacI-GFP. Student's t-test was performed to determine significance. (**D**) Anaphase TetR-mCherry intensities for the indicated strains, synchronized in G1 by alpha-factor treatment and released at the indicated temperatures. Intensities for G1 were determined 5 min after release from alpha-factor induced arrest. All data are means and standard deviation for n>30 cells. **** p<0.0001 and n.s. not significant.

observed a similar, and significant, decrease in reporter brightness in anaphase compared with G1 cells at this locus (*Figure 1C*). As for the *TRP1* locus, mutants in the H3 S10 pathway also affected fluctuations in intensity at the CEN distal locus: *hst2Δ*, *H3 S10A* and *H3S10D hst2Δ* showed a continuously higher fluorescent intensity on the LacO repeats near the *LYS4* gene and the *H3 S10D* mutation also resulted in a continuously lower fluorescent signal. These data indicated that changes in chromatin organization during mitosis indeed affected either the recruitment or the fluorescence intensity of TetR-mCherry and LacI-GFP on two distant chromatin loci, one close to the centromere and the second in the middle of the second longest yeast chromosome arm.

Since chromatin condensation is regulated by the kinase aurora B (Ipl1 in budding yeast), we last asked whether Ipl1 activity is required for the intensity decrease of TetR-mCherry at the *TRP1* locus in mitotic cells. We arrested wild type yeast cells and cells containing the temperature sensitive *ipl1-321* allele in G1 with alpha-factor, released them at the restrictive temperature of 35°C and determined TetO/TetR-mCherry fluorescence intensity in the same G1 or following anaphase (*Figure 1D*). Whereas wild type cells showed no significant difference in G1 and anaphase TetO/TetR-mCherry fluorescence intensity, the *ipl1-321* strain showed a significantly brighter dot when undergoing

anaphase at the restrictive temperature. Compaction in G1 of *ipl1-321* cells at the restrictive temperature was not affected, presumably due to the fact that this protein has no activity in G1, even in wild type cells (*Buvelot et al., 2003*). Thus, we conclude that the enhanced H2A-H4 interaction triggered by aurora B-dependent recruitment of the deacetylase Hst2 onto chromatin indeed affects the intensity of the TetR-mCherry signal on the chromosome.

## Fluorophore concentration quenching causes fluctuations in brightness

We next wanted to better understand the molecular processes and structural changes of chromatin that were underlying the fluorescence variation at the TetO array over the cell cycle. Assuming enhanced nucleosome-nucleosome interaction promotes chromatin compaction, three models may explain the observed decrease of fluorescence in mitosis. First, chromatin compaction might reduce access of DNA-binding proteins, such as TetR-mCherry, to their binding site on DNA and cause their removal, as postulated by the chromosome cleansing hypothesis. Second, chromatin compaction might increase the local packing of TetR-mCherry, leading to quenching of the fluorophore (*Lakowicz, 2013*); these two first models are depicted in *Figure 2A*. Third, the changed local environment of mitotic chromatin might reduce the intrinsic fluorescence of mCherry and GFP.

In order to better distinguish between these models, we rationalized that coexpressing TetR-GFP with TetR-mCherry would not protect mCherry from a cleansing effect (model 1) or a change in local environment (model 3), but should strongly reduce any quenching, due to intercalation of a second fluorophore with a different excitation spectrum. Furthermore, in this context, quenching might be replaced by Förster Resonance Energy Transfer (FRET) between the TetR-GFP and TetR-mCherry molecules. Remarkably, unlike the cells expressing only TetR-mCherry, cells expressing both versions of TetR failed to show significant variation of the fluorescence signal for either mCherry or GFP at the TetO array between anaphase and G1 (*Figure 2B*). Thus, cleansing and a general change in the local environment of the fluorophores are unlikely to explain the fluorescence drop observed at the TetO array during anaphase in the cells expressing solely TetR-mCherry. Supporting the idea that the intensity drop was due to a quenching effect, FRET was indeed observed upon exciting in the GFP and recording emission in the red channel in cells expressing both TetR-mCherry and TetR-GFP, but not in cells expressing TetR-mCherry alone (*Figure 2C*). Moreover, FRET was significantly increased during anaphase compared with G1-phase (*Figure 2B,C*), indicating that the fluorophores are indeed brought in closer proximity during anaphase compared with interphase. We conclude that increased H2A/H4 interaction results in a tighter packing of fluorophores and their quenching, establishing that H2A/H4 interaction leads to compaction of mitotic chromatin in vivo. Furthermore, cell cycle dependent changes of TetR-mCherry or TetR-GFP signals on TetO arrays is a reliable measure of short-range compaction of the underlying chromatin.

## Chromatin compaction precedes the axial shortening of chromosomes

Next, we investigated the dynamics of chromatin compaction during the cell cycle. To this end, we visualized both the TetO/TetR-mCherry (at *TRP1*) and LacO/LacI-GFP (at *LYS4*) loci simultaneously.This presence of two labeled loci on the same chromosome allowed measuring the physical distanceseparating them and hence the long-range contraction of the chromosome arm along its longitudinal axis during anaphase. Using this strain, we first recorded time-lapse movies (*Figure 3A*) in which we measured the intensity of the LacI-GFP fluorescence at the *LYS4* locus in cells progressing through mitosis (*Figure 3B*). Upon averaging the signal of at least 15 (t = -18 minutes) to maximum 31 (t = 0 minutes) such traces, we observed that the intensity of the signal was indeed lowest during the first 12 minutes of anaphase, while starting to increase as soon as the cells started to exit mitosis (*Figure 3B*, blue line indicates the formation of the first bud in the population). We also noticed that fluorescence intensity at the LacO locus was highly variable throughout every single movie, leading to high standard deviations. This variation was lowest during anaphase and started to increase as soon as chromatin was decondensing, consistent with the idea that chromatin is more constrained when it is most compacted and fluorophore quenching is highest. The source of this fluorescence variation is not known, but might reflect breathing movements of the underlying chromatin or complex photochemistry effects. In either case, this intrinsic cell-to-cell variability precludes drawing conclusions at the single cell level and emphasizes the fact that the quenching assay introduced here is statistical in nature.

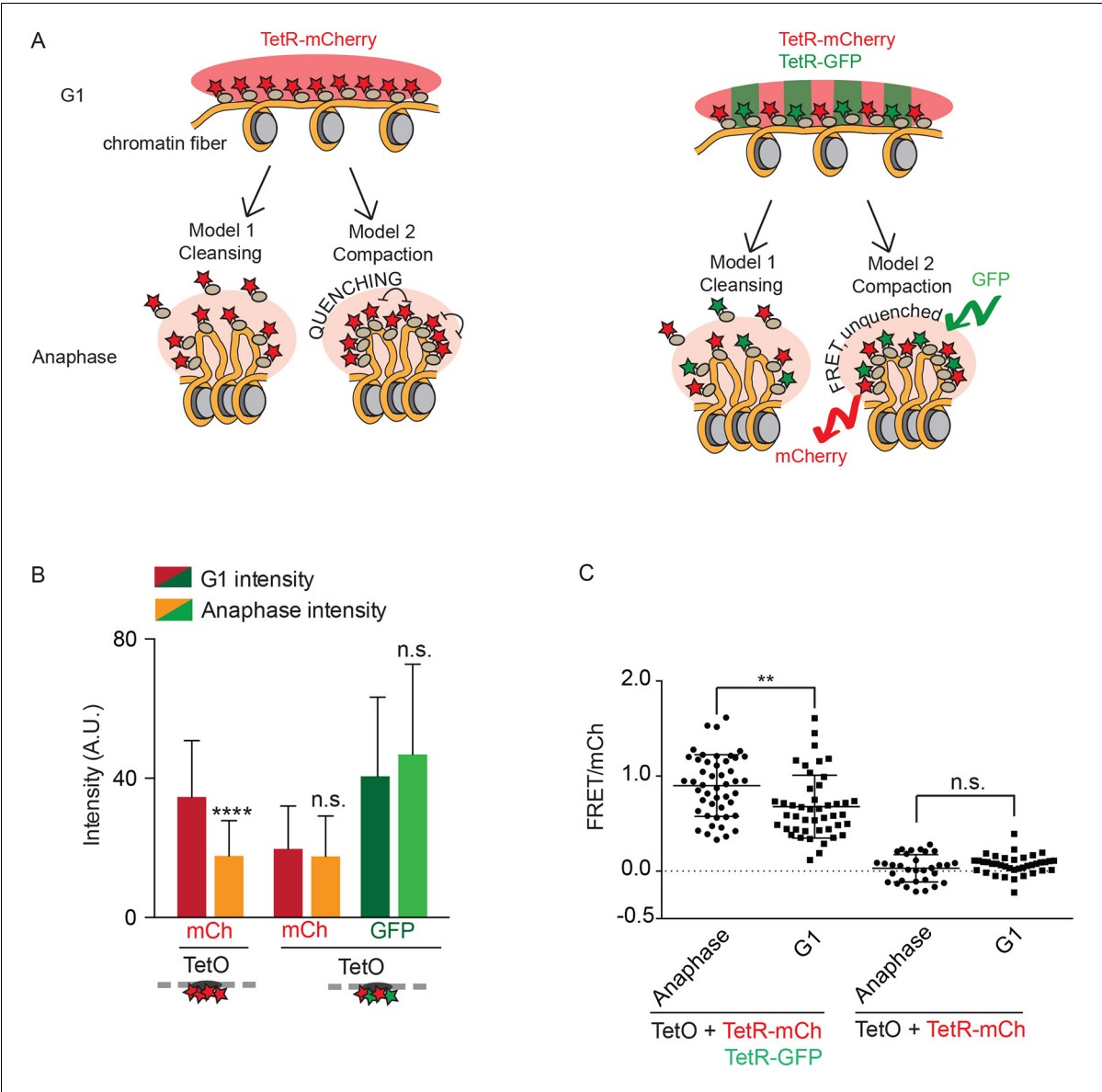

**Figure 2.** Fluorophore quenching causes changes of TetO/TetR intensity over the cell cycle. (**A**) Two models to explain anaphase-specific decrease in fluorescence brightness (cleansing and quenching, see text for explanations). Shown are the consequences of each model in G1 and anaphase, in the case of cells carrying TRP1:TetOs and either expressing only TetR-mCherry or TetR-mCherry and TetR-GFP. (**B**) G1 and anaphase TetO/TetR-mCherry intensities in cells carrying TetO and expressing only TetR-mCherry (left) or TetR-mCherry and TetR-GFP (right). Data are means and standard deviations, unpaired Student's t tests were performed to test significance, **** p<0.0001 and n.s. not significant. (**C**) FRET values for indicated strains. Plotted are mean values and standard deviation. Unpaired Student's t tests were performed to test significance, ** p<0.01 and n.s. not significant.

Next, we wanted to determine whether the dynamics of chromatin compaction could be related to the contraction of the chromosome arm measured using the *TRP1-LYS4* distance, as described previously by us and others (*Neurohr et al., 2011*; *Petrova et al., 2013*; *Guacci et al., 1994*; *Vas et al., 2007*). To avoid variations due to photobleaching, we used snapshot images of cells at precise and representative time points in mitosis (*Figure 3C*): metaphase (large buds but neither the *TRP1* nor the *LYS4* loci were separated), early anaphase (sister *TRP1* loci – in red – are separated, but the two *LYS4* loci are not), mid-anaphase (both loci have undergone separation but the *LYS4* locus still lags behind), late anaphase (all loci are separated and moved to the opposite poles of the cell) and G1 (unbudded cell). For each of these stages (>25 cells each), we measured both the intensity of the fluorescence on the two arrays and the distance between them. In this

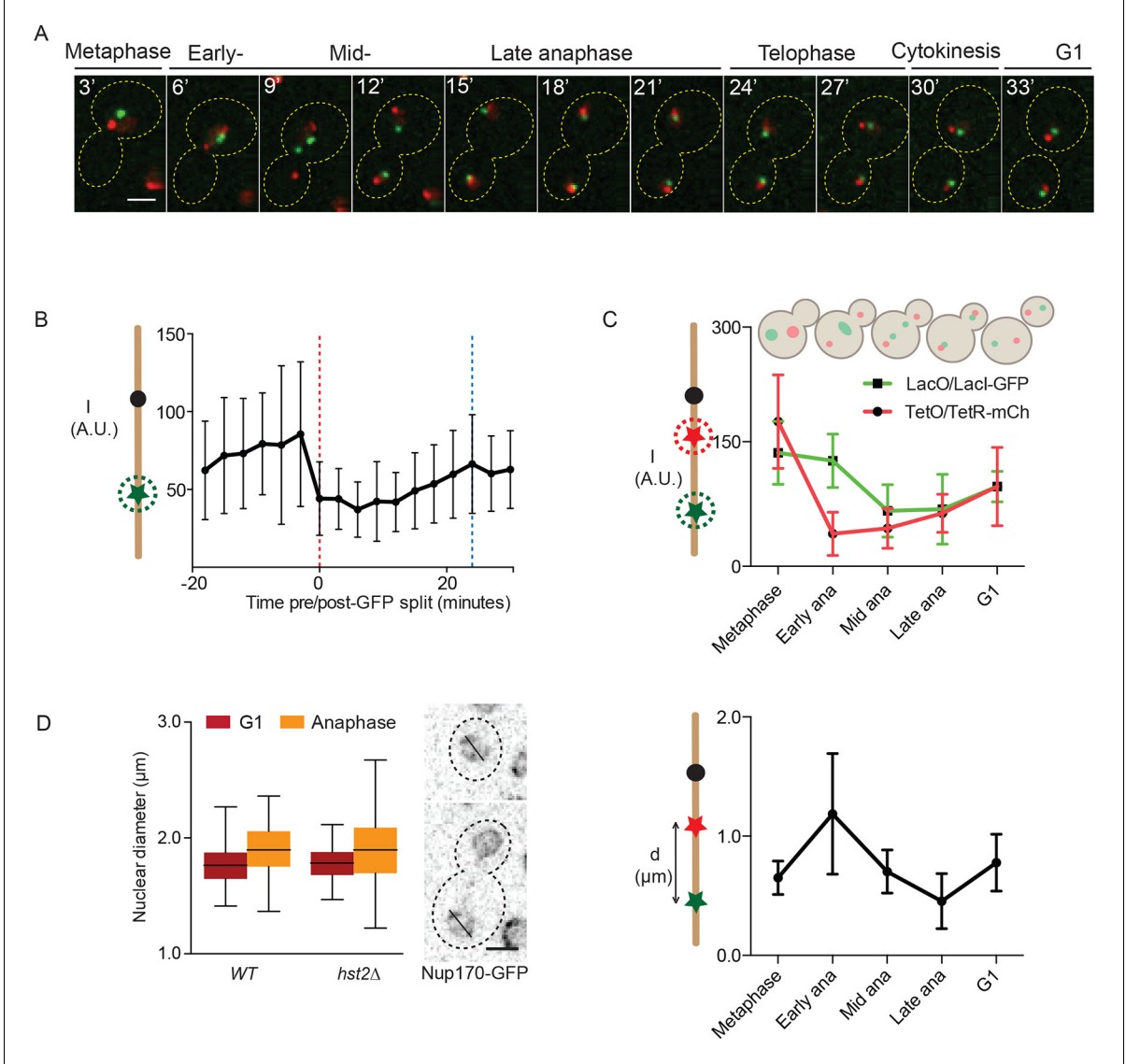

**Figure 3.** Dynamics of chromatin compaction and chromosome arm contraction. (**A**) Example of a cell going from metaphase to the next G1 phase with *TRP1* and *LYS4* loci marked with TetR-mCherry and LacI-GFP, respectively. (**B**) Background normalized, mean GFP-intensity values of mitotic cells, aligned at mid-anaphase (red dashed line: GFP dot split). Blue line indicates formation of the first bud. Standard deviations are shown. (**C**) Upper panel: normalized (to G1) intensity of TetR-mCherry and LacI-GFP foci in mother cells in the indicated cell cycle stages. Lower panel: mother *TRP1:TetO - LYS4:LacO* distances in indicated cell cycle stages. Shown are mean and standard deviation for n>30 cells. (**D**) Nuclear diameter of G1 and late anaphase cells in wild type and *hst2Δ* cells containing Nup170-GFP. Box shows median value, whiskers all data points n>50 cells. Scale bars are 2 μm.

study, we focused specifically on the loci segregated to the mother cell, as we showed before that mother and bud are not directly comparable (*Neurohr et al., 2011*).

Analysis of this data set indicated that the two marked loci underwent compaction and decompaction with slightly different kinetics (*Figure 3C*, upper panel). During early anaphase, both the TetO and LacO array seemed to be compacted to some extent already. As cells progressed to mid anaphase, the *CEN4* proximal TetO arrays seemed slightly more compacted than the distal LacO arrays. In late anaphase, the TetO array was already starting to unpack, whereas the LacO remained compacted. Both dots had recovered their full intensity in the G1 cells, demonstrating that the intensity decrease in anaphase was not solely the consequence of sister-chromatid separation, which is expected to reduce fluorescence intensity by half, down to its G1 level until the next S-phase. In the same cells, measuring the TetO-LacO distance (*Figure 3C*, lower panel) indicated that

the chromosome first stretched out upon anaphase onset, to subsequently contract, reaching their shortest length in late anaphase, as reported (*Guacci et al., 1994*; *Harrison et al., 2009*). The TetO-LacO distance re-extended then to its steady state average in G1 cells.

The changes in distance between the two loci could be due to changes in nuclear diameter, which would constrain the maximal distance that the loci can move apart by random motion. However, we did not observe any significant changes in nuclear diameter (as determined by GFP-tagging the nucleoporin Nup170) when comparing late anaphase and the G1 phase in wild type and *hst2Δ* cells (*Figure 3D*). Thus, the shortening of the *TRP1-LYS4* distance in late anaphase cells truly reflects the effect of chromatin condensation by axial contraction of the chromosomes. Furthermore, our results established that short-range chromatin compaction was not strictly concomitant with long-range axial contraction of the chromosome, but rather preceded it.

## Condensin is not involved in local chromatin compaction

These observations suggested that chromatin compaction and axial contraction of mitotic chromosomes might be distinct processes. Thus, we asked whether condensin, which is essential for axial chromosome contraction, contributed to short-range compaction of chromatin. We analyzed the brightness of the TetR-mCherry focus in yeast cells carrying the *smc2-8* allele, a temperature sensitive mutation in the condensin subunit Smc2 (*Figure 4A*). Remarkably, condensin inactivation for 90 min at the restrictive temperature had no effect on the changes in mCherry brightness between the anaphase and G1-phase of the cell cycle, whereas it indeed abrogated shortening of the TetO-LacO distance during anaphase (see below). We therefore conclude that condensin, unlike histone 3 phosphorylation and histone 4 deacetylation, does not promote nucleosome-nucleosome interaction. To test this idea further, we directly probed H2A/H4 interaction by using genetically encoded Ultraviolet (UV) inducible crosslinking (*Wilkins et al., 2014*). We arrested cells in G1 with alpha-factor and released them in the presence of nocadozole under wild type and *smc2-8* conditions at 37°C. Fluorescence-Activated Cell Sorting (FACS) analysis showed that the release and arrest was equally efficient in both cells (*Figure 4—figure supplement 1*). In wild type cells, as reported before, H4/H2A crosslinking is observed in mitosis and correlated strongly with H4 K16 deacetylation (*Figure 4B*). In fitting with the microscopy data (*Figure 4A*), the crosslinking between H4 and H2A showed no difference in kinetics in the condensin inactivated and in the wild type cells (*Figure 4B*). Thus, condensin function is not required for proper, short-range compaction of mitotic chromatin.

## The chromatin compaction pathway does not contribute to axial contraction of chromosomes

In order to investigate in more detail how the phosphorylation of H3 S10 and the subsequent activation of Hst2 contributed to chromosome condensation, we next characterized if these events promoted axial contraction of chromosome IV, using the LacO-TetO distance as a readout (see *Figure 3D*). Confirming the role of aurora B/Ipl1 in chromosome condensation, the *ipl1-321* mutant cells failed to undergo chromosome contraction when shifted to the restrictive temperature prior to mitosis, compared with wild type cells at these temperatures (*Figure 5A*). As expected from previous studies (*Neurohr et al., 2011*; *Lavoie et al., 2002*), the function of Ipl1 in the contraction of regular chromosomes was unlikely to require H3 S10 phosphorylation and H2/H4 interaction, since the mutations *H3 S10A* and *H4 Δ9–16* did not impair anaphase contraction (*Figure 5B,C* ). Thus, Ipl1 promotes the axial contraction of chromosomes independently of phosphorylating H3 S10 and of promoting H4/H2A interaction, but possibly by promoting condensin function (see discussion).

In contrast, the *hst2Δ* mutation did abrogate the proper contraction of the chromosome during mitosis, implying that Hst2 acts in both axial chromosome contraction and short-range chromatin compaction (*Figures 5C* and *1B*). Even more remarkably, the *H3 S10D* phospho-mimicking allele caused chromosome IV to remain in a constitutive state of axial contraction throughout the cell cycle (*Figure 5B,C*). This most probably reflected constitutive recruitment of Hst2 to nucleosomes, since inactivation of the *HST2* gene in these cells abolished contraction (*Figure 5B,C*). Thus, boosting the recruitment of Hst2 to chromatin through mimicking constitutive and ubiquitous phosphorylation of H3 promoted both chromatin compaction and axial chromosome contraction, despite the fact that the H3-dependent pathway of Hst2 recruitment is normally dispensable for Hst2 function in axial chromosome contraction during regular mitoses (though, it is essential for adaptive hyper-

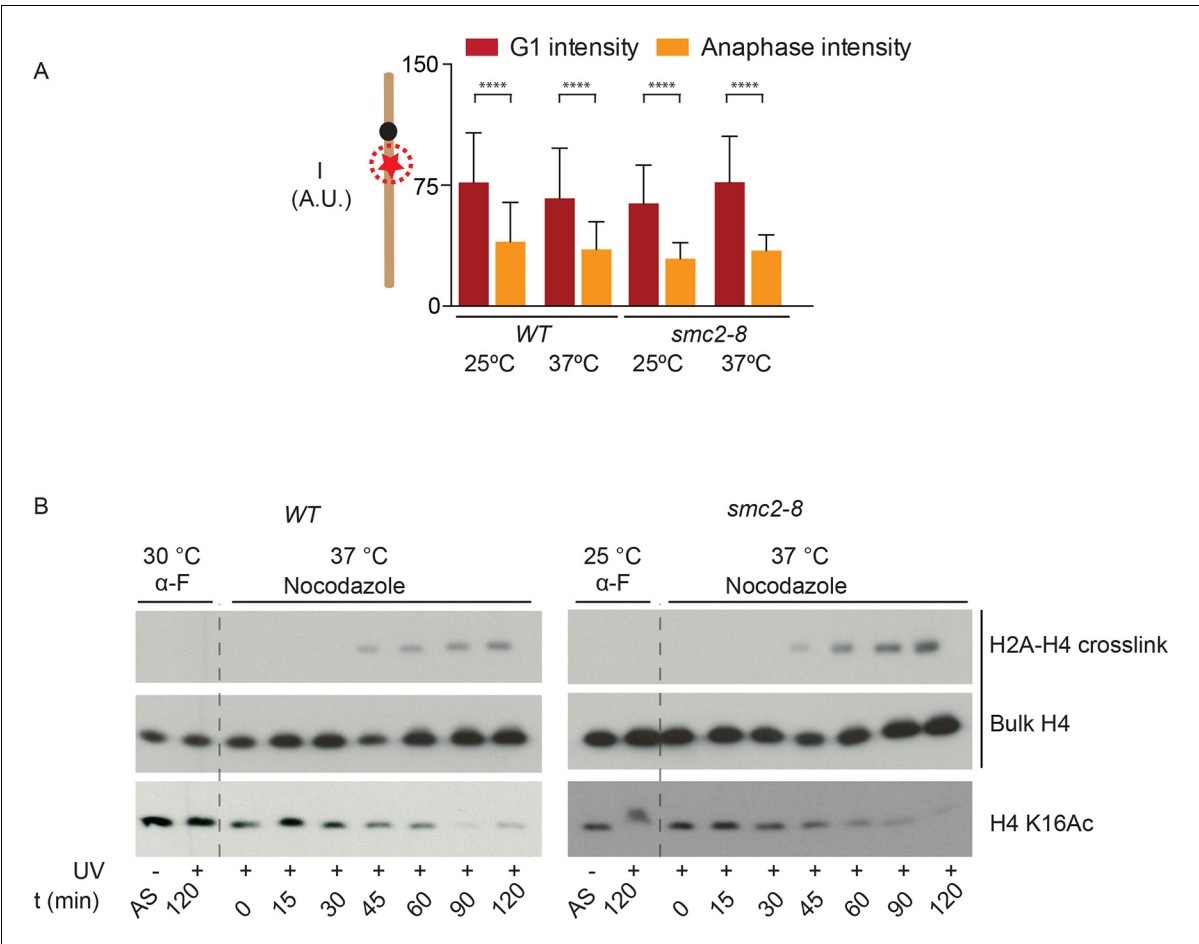

**Figure 4.** Condensin does not impact chromatin compaction. (**A**) TetR-mCherry intensities in the mother cell for the indicated strains and cell cycle stages. To inactivate Smc2, cells were shifted to 37°C for 90 min. One way ANOVA was performed to test significance, **** p<0.0001 and n>40. (**B**) Yeast cells producing H2A Y58BPA were synchronized with alpha-factor at permissive temperature and then released into medium containing nocodazole at restrictive temperature. Samples were taken at indicated times, irradiated with UV and histones extracted with acid from isolated nuclei. Western blot against H4 detects the H2A-H4 crosslink (upper row), bulk H4 (lower row) and blotting against H4 K16Ac shows cell cycle progression.

The following figure supplement is available for figure 4:

**Figure supplement 1.** FACS analysis of alpha-factor synchronized cells released into nocodazole (wild type or *smc2-8*).

condensation of an artificially long chromosome). We next sought to determine the role of H4 K16 deacetylation in anaphase chromosome contraction. In accordance with the idea that H4 deacetylation solely plays a role in mitotic chromatin compaction, the *H4 K16R* allele did not show any significant differences in chromosome contraction compared with wild type cells (*Figure 5D*). When we deleted *HST2* on top of *H4 K16R*, we observed a significant decrease in chromosome contraction as compared with wild type and *H4 K16R* cells, but no significant difference with *hst2Δ* cells (*Figure 5D*). Based on this we conclude that H4 deacetylation has no evident role in chromosome contraction. Furthermore, our data suggest that *HST2*, when deleted, does not have its defects on chromosome contraction through the chromatin compaction pathway, but rather by acting on another factor.

Together, in agreement with the fact that they follow different kinetics (*Figure 3*), our data reveal that axial contraction of chromosomes and compaction of chromatin are two independent processes. They depend on distinct molecular pathways that are at least in part coordinated by the same regulatory input, namely the activity of the kinase aurora B and the deacetylase Hst2.

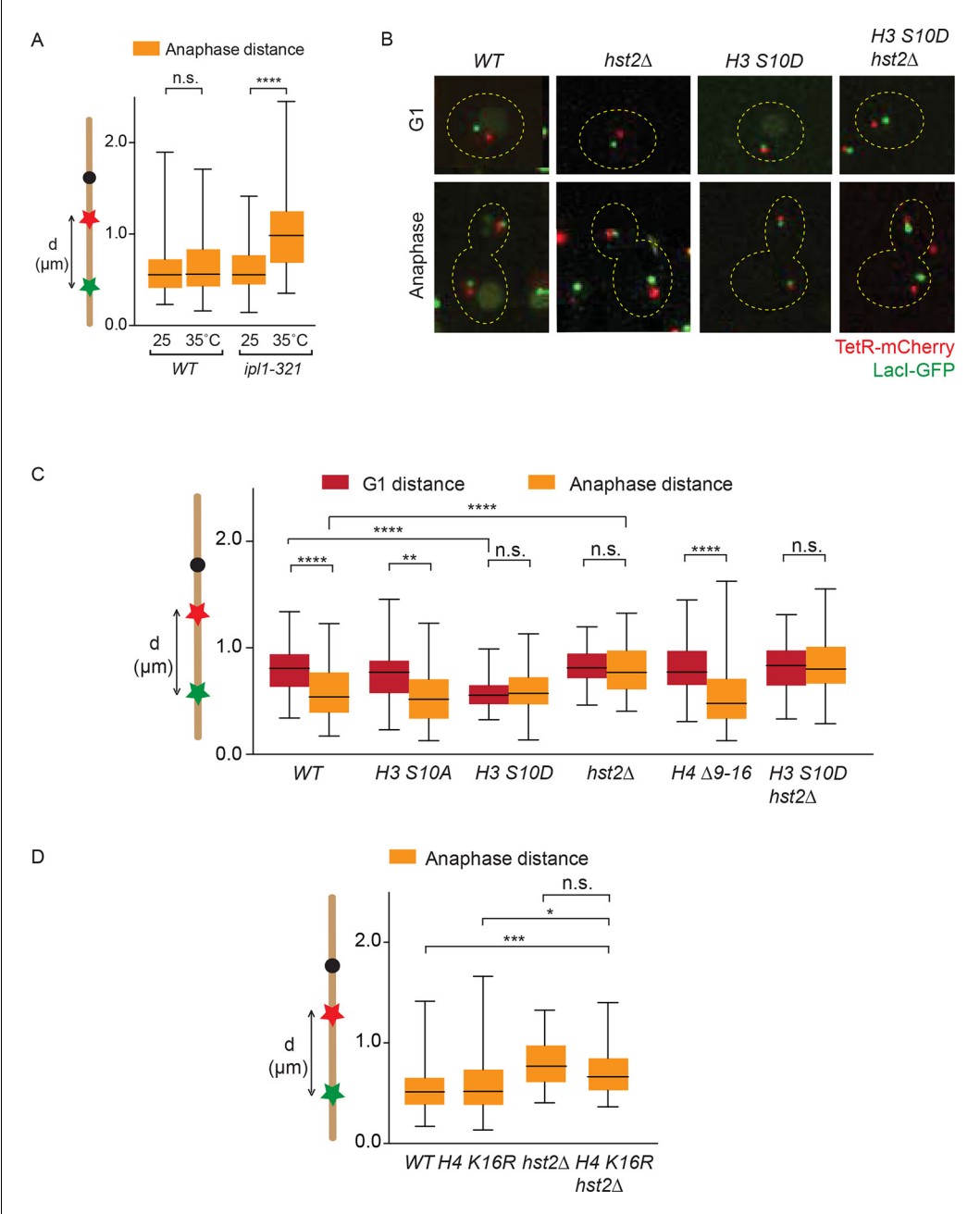

**Figure 5.** Chromatin compaction does not influence axial chromosome contraction. (A) *TRP1-LYS4* distances for the indicated strains, synchronized in G1 by alpha-factor treatment and released at the indicated temperatures. (B) *TRP1-LYS4* distances were determined in the mother cell for the indicated strains and cell cycle stages. Box shows median value, whiskers all data points n>30 cells. One way ANOVA was performed to test significances, ** p<0.01, **** p<0.0001, n.s. not significant. (C) Example cells containing the indicated mutations and their impact on chromosome length as determined by the *TRP1* (red) to *LYS4* (green) distance. (D) *TRP1-LYS4* distances were determined for the indicated strains in anaphase. Box shows median value, whiskers all data points n>45. One way ANOVA was performed to test significances, *** p<0.001, * p<0.05, n.s. not significant.

## Hst2 may promote condensin function

The data above demonstrated that *hst2Δ* mutant cells have strong defects in axial chromosome contraction, while other components of the chromatin compaction pathway (H3 S10 and H4) do not. Thus, we rationalized that Hst2 might not only promote chromosome compaction, but also the function of factors required for the axial contraction of chromatids, such as condensin. To

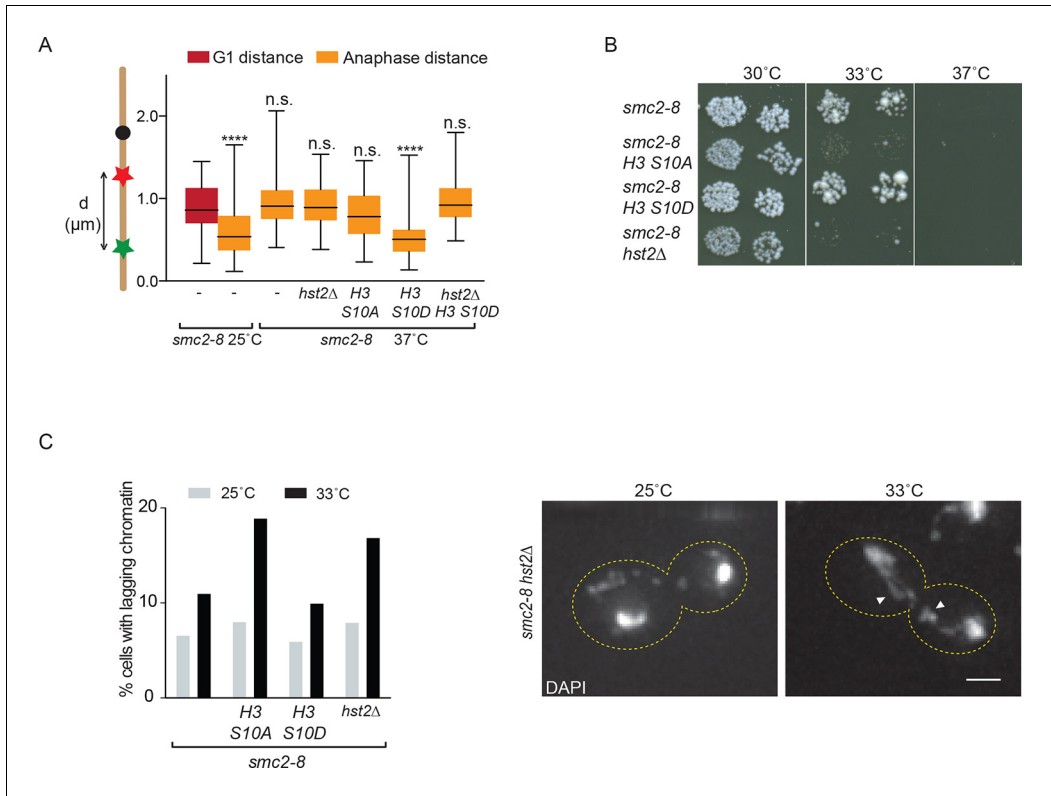

**Figure 6.** Condensin and *HST2* are in the same genetic pathway that ensures proper chromosome segregation. (**A**) *TRP1-LYS4* distances were determined for the indicated strains in anaphase after shifting cells for 90 min to 37°C. Box shows median value, whiskers all data points n>30. One way ANOVA was performed to test significance between G1 in *smc2-8* at 25°C and other strains, **** p<0.0001 and n.s. not significant. (**B**) Spotting assay of indicated strains on YPD plates at the indicated temperatures. (**C**) Percentage of anaphase cells containing anaphase bridges for the indicated strains after 90 min at 25°C or 90 min at 33°C. DAPI, 4', 6-diamidino-2-phenylindole; YPD, yeast extract peptone dextrose. n>240, scale bar is 2 µm.
The following figure supplement is available for figure 6:

**Figure supplement 1.** Plate growing spotting assay on YPD plates of wild type, *H3 S10A, H3S10D* and *hst2Δ* strains at different temperatures.

test this notion, we measured the effect of combining the *smc2-8* mutation with *H3* and *hst2Δ* mutations on axial chromosome contraction during anaphase. As expected, the *smc2-8* single mutant cells grown at the restrictive temperature failed to axially contract their mitotic chromosomes to wild type levels (*Figure 6A*). Deletion of Hst2 in these cells did not exacerbate their contraction phenotype (*Figure 6A*), suggesting that condensin and Hst2 might act in the same genetic pathway. In contrast, contraction was restored in the *H3 S10D smc2-8* double mutant cells. This effect was likely due to boosting Hst2 recruitment and activity since it disappeared in the *H3 S10D hst2Δ smc2-8* triple mutant cells. The effect of H3 most likely depended on the phosphorylation state of S10, since the *S10A* mutation did not promote contraction. Thus, enhanced activation of Hst2 (*H3 S10D*) suppressed the effect of the *smc2-8* mutation on chromatin contraction. Hst2 might mediate the suppression of the contraction defect due to the *smc2-8* mutation either through activation of a still unknown, alternative contraction pathway or through stimulation of condensin activity such as to restore at least part of its function.

# Axial contraction and chromatin compaction synergistically facilitate chromosome segregation

In order to establish the physiological relevance of our phenotypic observations, we next asked whether combining compaction and contraction defects cell growth and viability. Consistent with condensin and H3 S10 phosphorylation pathways acting synergistically to promote proper chromosome condensation, the *smc2-8 H3-S10A* double mutant cells were growing slower than the *smc2-8* single mutant cells, and were unable to grow at 33°C, unlike the single mutant cells (*Figure 6B* and *Figure 6—figure supplement 1*). Accordingly, the *smc2-8 hst2Δ* double mutants showed the same phenotype. Thus, when the condensin complex was only partially active, cells depended on Hst2 and phosphorylation of histone H3 on S10 for growth. Remarkably, however, the *smc2-8 H3 S10D* double mutant did not show any altered or improved growth or viability phenotype compared with the *smc2-8* single mutant cells. It grew reasonably well at 33°C, while being non-viable at 37°C. Thus, restoring axial contraction of the chromosomes using the *H3 S10D* mutation was not sufficient to restore the viability of the cells lacking condensin function. These results are consistent with condensin playing more roles than simply promoting the axial contraction of chromosomes.

To further test the requirement of condensin and H3 S10 pathways on chromosome segregation, we imaged anaphase cells stained with 4', 6-diamidino-2-phenylindole (DAPI) to visualize the frequency with which the single and double mutant strains produced lagging chromatin. The cells were grown at 25°C, the permissive temperature for the *smc2-8* mutation or shifted for 90 min to 33°C, a semi-permissive temperature for *smc2-8* mutant cells. At 25°C, the *smc2-8* single and the *smc2-8 H3 S10A*, *smc2-8 hst2Δ* and *smc2-8 H3 S10D* double mutants cells showed no strong differences (6.3–8.0%) in the frequency at which lagging chromatin was observed in the center of the spindle (*Figure 6C*). In contrast, the *smc2-8 H3 S10A* and the *smc2-8 hst2Δ* double mutants grown at 33°C showed a marked increase in the frequency (18.9% and 16.9%, respectively) of lagging chromatin compared with the *smc2-8* single mutant cells. The *smc2-8 H3 S10D* mutant cells did not show such an additive phenotype (10.0%; *Figure 6C*). These data indicate that the pathways promoting the short-range compaction of chromatin and the axial contraction of chromosomes contribute synergistically to shaping mitotic chromosomes in order to ensure their correct segregation prior to cell division.

## Discussion

Mitotic chromosome condensation is the process through which a relatively relaxed interphase chromosome condenses into two relatively short, compact sister chromatids that the spindle can symmetrically segregate. The kinase aurora B has long been implicated in this process (*Lavoie et al., 2004*; *Tada et al., 2011*; *Lavoie et al., 2002*). Through activation of condensin and phosphorylation of S10 on histone H3 it is thought to promote the formation of a mitotic chromosome. However, how these pathways interacted with each other to shape mitotic chromosomes was poorly understood.

Here, we monitored chromosome condensation in living yeast cells by using two microscopy assays, allowing us to distinguish long-range contraction of chromosomes along their longitudinal axis from short-range chromatin compaction. The first assay has been used before (*Petrova et al., 2013*; *Guacci et al., 1994*; *Vas et al., 2007*), but did not detect a role for histone phosphorylation during regular mitoses (*Neurohr et al., 2011*). The second assay, introduced here, uses fluorophore quenching to show that nucleosome-nucleosome interaction via H3 phosphorylation and in an Hst2-dependent manner promotes short-range compaction of mitotic chromatin in vivo. This enhanced packing is evidenced by the increased quenching of fluorophores when the chromosome locus is decorated by multiple copies of a fluorescent reporter. The quenching effect was observed similarly well whether we used GFP or mCherry as a fluorophore, and independently of whether it was fused to TetR or to LacI. Though we cannot exclude any local effects of the repetitive nature of the TetO and LacO repeats on chromatin compaction, this novel fluorescence-based assay allows for a simple detection of chromatin compaction states. Beyond promoting chromosome segregation, the exact function of this compaction process will be interesting to address. Our studies establish that chromatin compaction does not displace DNA-associated proteins such as TetR or LacI, as a strong cleansing model would predict. Although they do not exclude that other DNA-binding factors might be removed from the DNA during this process, these data suggest that compaction might serve other

functions, such as to change the biophysical properties of the chromatin fiber during segregation. Chromatin compaction might for example affect the local stiffness, elasticity and mechanical resistance of chromosomes to facilitate their decatenation, protection and movement during anaphase.

Related to this, it is remarkable that the *H3 S10D* mutation is able to establish an Hst2-dependent and constitutive state of compaction and contraction, as evidenced by both assays, without affecting much growth and viability. We propose that this mutation strongly boosts the recruitment and activation of Hst2 all along chromosomes by mimicking the phosphorylated state of histone H3. Two models may explain how *H3 S10D* elicits its effects. In the first, *H3 S10D* might lead to hyper-activation of chromatin compaction by over-recruitment of Hst2, such that it might shorten the chromosome axis, independently of the contraction machinery. However, the following observations speak against this idea: I. crosslinking studies do not indicate that *H3 S10D* increases nucleosome-nucleosome interaction (*Wilkins et al., 2014*), II. fluorescence quenching of TetR-mCherry and LacI-GFP is not enhanced in the *H3 S10D* mutant (*Figure 1B,C*) compared with WT anaphase cells, and III. phosphorylation of H3 S10 does promote the condensin-dependent and Hst2-dependent contraction of artificially long chromosomes (*Neurohr et al., 2011*; *Wilkins et al., 2014*). Thus, these data support an alternative model: Hst2 may stimulate condensin function, and hyper-activation of Hst2 by *H3 S10D* may promote this effect and thereby chromosome arm contraction. To fully distinguish between these two models it will be important to investigate the effect of the *H3 S10D* mutation on condensin loading and activity. Furthermore, it is important to note that the phenotype of the *H3 S10D* mutant cells does not seem to reflect physiological conditions taking place during regular mitoses, indicating that the fraction of Hst2 driving wild type chromosome contraction does not depend on H3 S10 phosphorylation (dashed arrow *Figure 7*, left panel). Rather, it might reflect what happens when both chromosome condensation pathways are hyper-activated, for example under the presence of an artificially long chromosome or in small cells, such as cells grown on a poor carbon source (see below). In any case, it is striking that the *H3 S10D* strain does not show any obvious defects in growth and viability despite causing chromatin compaction and contraction throughout the cell cycle. Thus, this constitutively condensed state does not impair access and remodeling of chromatin by the transcription and replication machineries during interphase in any major manner.

Our kinetic data suggest that chromatin undergoes a number of successive structural changes during anaphase and that compaction precedes contraction. In this respect, two models could explain the extended *TRP1-LYS4* distance observed in early anaphase cells. First, this might simply be due to the fact that the anaphase nucleus is more elongated, allowing for the chromosome arm to reach its real extension without being confined in the normally spherical morphology of late anaphase or G1 nuclei. In this scenario, the apparent stretching of the chromosome in early anaphase does not correspond to any rearrangement of the chromatin or changes in contraction, and the impression that contraction follows compaction is an illusion. However, previous work has established that residual cohesion between sister chromatids persists during early anaphase and that this actually causes chromosome stretching during early segregation stages (*Renshaw et al., 2010*;

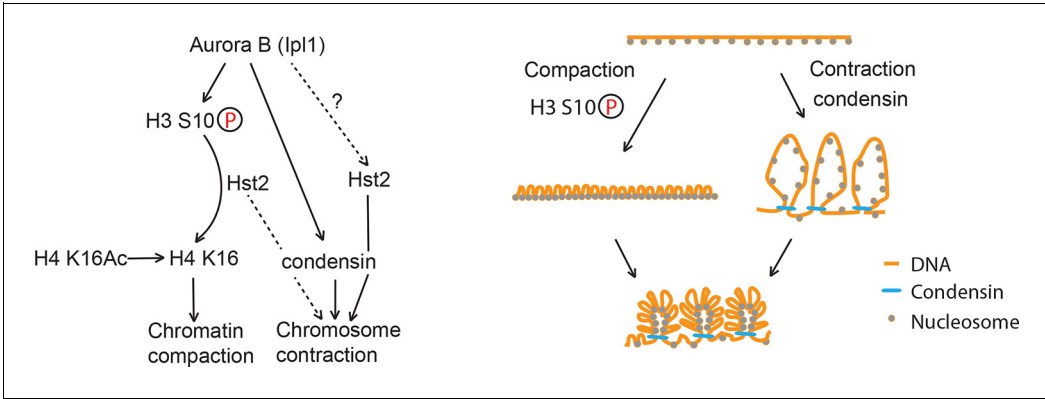

**Figure 7.** Model of chromosome condensation. H3 S10 phosphorylation leads to chromatin fiber closure, whereas condensin is required for axial shortening. Hst2 is needed for both of these levels of condensation.

*Harrison et al., 2009*). Thus, a second model would be that maximum contraction is only reached when cohesion is fully resolved, such as to not interfere with cohesion removal. This scenario suggests that chromosome contraction indeed needs to be a late anaphase event to complete chromosome segregation, and that perhaps the compaction events that precede play an important role in facilitating cohesion resolution. This model is attractive because it might more thoroughly account for the molecular events that are actually taking place during anaphase. Understanding how these processes are regulated during early anaphase will be necessary in order to distinguish between the two models. However, one definite conclusion that we can already draw is that arm contraction and chromatin compaction undergo relaxation between late anaphase and G1, since these nuclei are not significantly different in size, and hence the effects observed are not due to confinement alone.

Based on our data on chromosome contraction and compaction in *Figure 3C*, we propose that mitotic chromosome condensation entails at least three processes (*Figure 7*).

First, a histone 3-dependent and histone 4-dependent process, which we term chromatin compaction, ensures the short-range tightening of DNA into a smaller volume via nucleosome-nucleosome interaction (*Wilkins et al., 2014*). This process strictly depends on the phosphorylation of histone H3 on serine 10, and the subsequent deacetylation of lysine 16 of histone H4. Because this event has little impact on the length of condensed chromosomes, it has escaped attention until now. However, our data indicate that it is an important event for proper chromosome architecture and is an early event of every yeast mitosis.

Second, a condensin-dependent process, termed here axial chromosome contraction, ensures the long-range contraction of the chromosome, probably by facilitating the formation of higher order chromatin structures along the chromosome axis (*Cuylen and Haering, 2011*; *Baxter and Aragón, 2012*). This process is completed after chromatin compaction, in late anaphase, but does not strictly require it in order to proceed. Certainly, it will be interesting to clarify this order of event, to dissect how it is controlled, and to determine whether it facilitates the ordered condensation of well-separated chromosome arms. Moreover, the observation that Hst2 contributes to both processes suggests that this enzyme might be pivotal for coordinating compaction and contraction with each other. In this respect, it will be interesting to determine how Hst2 promotes the shortening of the chromosome axis, and whether it does so by directly modulating condensin function.

Third, our data also indicate the existence of an additional, also condensin-dependent process devoted to some other aspect of chromosome organization, beyond compaction and contraction. Indeed, restoring the contraction of the chromosome along its axis in the *smc2-8* condensin mutant cells by mimicking constitutive H3 phosphorylation did not alleviate the temperature-dependent lethality caused by the *smc2-8* mutation. Furthermore, although inactivation of the deacetylase Hst2 largely abolished the axial contraction of chromosomes, *HST2* is not an essential gene. Hence, we suggest that the essential function of condensin is not to mediate long-range contraction. Rather, the essential function of condensin most likely relates to its roles in orchestrating the decatenation of intertwined sister-chromatids (*Charbin et al., 2014*; *Baxter and Aragón, 2012*; *Baxter et al., 2011*) and the re-annealing of single stranded DNAs (*Sakai et al., 2003*).

In an earlier study we demonstrated that yeast cells challenged by the presence of an exceptionally long chromosome or by being themselves exceptionally small (for example, *whi3* mutant cells and cells growing on a poor carbon source) possess the ability to condense chromosomes beyond wild type levels to adjust the axial contraction of long chromosomes and adapt their length to the length of the spindle, a process we termed adaptive hyper-condensation (*Neurohr et al., 2011*), and which we would like to rename adaptive hyper-contraction. This process depended on condensin and phosphorylation of H3 S10, this last requirement being in contrast to what is observed for the contraction of most wild type chromosomes during regular mitoses. Accordingly, single *smc2-8* and *H3 S10A* mutants failed to hyper-contract an exceptionally long chromosome (*Neurohr et al., 2011*). Since the *H3 S10A* mutation had no defect on the axial contraction of wild type chromosomes, these results indicated that adaptive hyper-contraction depended more heavily on Ipl1-dependent phosphorylation of S10 on histone H3 than regular chromosome condensation does. Thus, we propose that H3 S10 is hyper-phosphorylated on long chromosomes by Ipl1 located in the center of the spindle (midzone), boosting Hst2 and condensin activity (dotted arrow, *Figure 7*). Our observation that mimicking S10 phosphorylation, using the *H3 S10D* allele, promotes axial shortening in an Hst2-dependent manner is indeed consistent with high levels of S10 phosphorylation promoting the crosstalk between compaction and contraction, via Hst2.

It will be interesting to test how the principles laid out here function in eukaryotes with a more complex chromosome structure, such as humans. A good starting point for such investigations would be to test the effects of H3 S10 mutation and/or removal of the Hst2 homologues on such cells. How important is a tight regulation between short-range chromatin compaction and long-range chromosome contraction? How does that impact chromosome structure? And how do such mutations affect the segregation of the genetic material? Together, our work demonstrates that yeast is a powerful system to dissect the mechanisms underlying chromosome condensation and how the disruption of such mechanisms affects chromosome segregation and cell viability.

## Materials and methods

### Yeast strains

All yeast strains used in this study were derived from the S288c background and are described in *Table 1*. Histone mutants were amplified from the synthetic non-essential histone collection from Thermo Scientific and transformed in S288c-derived strains. *HST2* was deleted by using standard methods (*Janke et al., 2004*). *smc2-8* temperature sensitive strains were obtained by crossing and tetrad dissection. Spotting assay in *Figure 5B* was done by aligning cells at OD600 = 0.1 and diluting 1:5 in subsequent steps. 2.5 µl drops were plated on yeast extract peptone dextrose (YPD) medium and grown for 2.5 days.

### Fluorescent microscopy

All strains were grown at 30°C on YPD medium. Condensin temperature-sensitive strains were grown in liquid YPD until $OD_{600}$ = 0.4 and then transferred to 25°C or 37°C for 90 min prior to imaging. For imaging, cells were resuspended in non-fluorescent medium (Synthetic Defined (SD) minus tryptophan (– TRP)) and put on an agar pad. All microscopy was done using a Deltavision microscope (Applied Precision) with a CCD HQ2 camera (Roper), 250W Xenon lamps, Softworx software (Applied Precision), and a temperature chamber set to the desired temperature. For both chromosome contraction and chromatin compaction assays, still images were taken using 750 ms exposure in the Tetramethylrhodamine (TRITC (red)) channel and 500 ms exposure time for the Fluorescein (FITC (green)) over ten Z-slices that were 0.5 µm apart. Transmission images were taken as a reference. Analysis was done using Fiji image processing software. We defined G1 cells as round cells with no bud and anaphase cells as a budded cell where the red focus leads and the green focus followed. All measurements were done in mother cells. Sum intensity projections were used and a line was drawn between the green and red foci to determine chromosome contraction. For fluorescence intensity, a Region Of Interest (ROI) was drawn around a focus and the integrated density was determined. An identically sized ROI was put in the nucleus to determine the background signal. Background intensity was subtracted from the focus intensity to yield the fluorescent intensity for a given focus. For the example cell in *Figure 3A*, 1-hr long movies were taken using slightly modified conditions: 300 ms FITC exposure, 500ms TRITC exposure and eight Z-stacks that were 0.5µm apart. FRET was done using the same microscope described above. First the FRET channel was recorded using 500 ms exposure in the FITC channel while capturing emission in the TRITC channel. Then, excitation and emission in TRITC was recorded using 500 ms of exposure time. Analysis was done by drawing a ROI around the fluorescent focus in both channels, subtracting the background and determining the ratio between them. At least 50 cells were measured for all conditions. For Ipl1 inactivation, exponentially growing cells were arrested with alpha-factor according to standard protocols and released at 25°C or 35°C. To obtain G1 condensation, cells were released for 5 min, to obtain anaphase condensation cells were released for approximately 2 hr at the indicated temperatures before imaging.

Nuclear diameters were determined by imaging Nup170-GFP as described above and by using Fiji image processing software.

DAPI staining was performed by growing cells of a given strain to $OD_{600}$ < 1.0 in YPD at 25°C, then two new diluted cultures were made at $OD_{600}$ = 0.2. One of them was again put at 25°C, while the other was kept at 33°C for 90 min. Cells were harvested by 90 s centrifugation in a 1.5 ml Eppendorf tube at 650 RCF. Cells were resuspended in 1 ml 70% ethanol, left to fix for 7 min, spun down again and washed in 1 ml sterile water, before being resuspended in 3–5 µl

**Table 1.** Yeast strains used in this study

| yYB number | Mating type | genotype |
|---|---|---|
| N/A | a | his3Δ1 leu2Δ0 met15Δ0 ura3Δ0 smc2::smc2-8:kanMX |
| N/A | a | his3Δ1 leu2Δ0 met15Δ0 ura3Δ0 |
| 3476 | a | trp1::TetO:TRP1 lys4::LacO:LEU2 his3::LacR-GFP:HIS3 TetR-mRFP |
| 3477 | a | trp1::TetO:TRP1 lys4::LacO:LEU2 his3::LacR-GFP:HIS3 TetR-mRFP ipl1-321 |
| 4699 | a | trp1::TetO:TRP1 lys4::LacO:LEU2 his3::LacR-GFP:HIS3 TetR-mRFP Spc42-GFP:hph hht1::hht1-S10A:KanMXloxP hht2::hht2-S10A:bleloxP ura3 ade1 leu2 |
| 8080 | a | Nup170-GFP:HIS3 ura3 ade1 leu2 |
| 9101 | a | his3Δ200 leu2Δ0 lys2Δ0 trp1Δ63 ura3Δ0 met15Δ0 can1::MFA1pr-HIS3 hht1-hhf1::NatMX4 hht2-hhf2::[HHTS-HHFS]-URA3 |
| 9332 | a | his3Δ200 leu2Δ0 lys2Δ0 trp1Δ63 ura3Δ0 met15Δ0 can1::MFA1pr-HIS3 hht1-hhf1::NatMX4 hht2-hhf2::[HHT-S10D-HHFS]-URA3 |
| 10818 | a | his3Δ200 leu2Δ0 lys2Δ0 trp1Δ63 ura3Δ0 met15Δ0 can1::MFA1pr-HIS3 hht1-hhf1::NatMX4 hht2-hhf2::[HHTS10AS]-URA3 |
| 11676 | a | smc2-8 hhf1-hht1::NatMX hst2::hphNT1 ura3 ade1 leu2 |
| 12002 | a | hst2::KanMX his3Δ1 leu2Δ0 ura3Δ0 met15Δ0 |
| 10116, 10117 | a | trp1::TetO:TRP1 lys4::LacO:LEU2 his3::LacR-GFP:HIS3 TetR-mRFP Hhf1-Hht1::NatMX hht2-hhf2::[HHT-HHFΔ9-16]-URA3 ura3 ade1 leu2 |
| 10122, 10123 | a | trp1::TetO:TRP1 lys4::LacO:LEU2 his3::LacR-GFP:HIS3 TetR-mRFP Hhf1-Hht1::NatMX hht2-hhf2::[HHT-HHFK16R]-URA3 ura3 ade1 leu2 |
| 10331 | a | Nup170-GFP:HIS3 hst2::NatMX ura3 ade1 leu2 |
| 11518, 11519 | diploid | trp1::TetO:TRP1/TRP1 lys4::LacO:LEU2/LYS4 TetR-mRFP leu2/leu2 ura3/ura3 his3/his3 |
| 11520, 11521 | diploid | trp1::TetO:TRP1/TRP1 TetR-mRFP/ TetR-GFP:LEU2 leu2/leu2 ura3/ura3 his3/his3 |
| 11173, 11989 | a | trp1::TetO:TRP1 lys4::LacO:LEU2 his3::LacR-GFP:HIS3 TetR-mRFP hht1::hht1-S10A:KanMXloxP hht2::hht2-S10A:bleloxP smc2-8 ura3 ade leu2 |
| 9782, 9783, 9784 | a | trp1::TetO:TRP1 lys4::LacO:LEU2 his3::LacR-GFP:HIS3 TetR-mRFP Hhf1-Hht1::NatMX ura3 ade1 leu2 |
| 10006, 10007, 10008 | a | trp1::TetO:TRP1 lys4::LacO:LEU2 his3::LacR-GFP:HIS3 TetR-mRFP Hhf1-Hht1::NatMX hst2::hphNT1 ura3 ade1 leu2 |
| 10274, 10276, 10277 | a | trp1::TetO:TRP1 lys4::LacO:LEU2 his3::LacR-GFP:HIS3 TetR-mRFP smc2-8 |
| 10450, 10451, 10452 | a | trp1::TetO:TRP1 lys4::LacO:LEU2 his3::LacR-GFP:HIS3 TetR-mRFP HHT2S10D:URA3 ura3 ade1 leu2 |
| 10584, 10585, 10586 | a | trp1::TetO:TRP1 lys4::LacO:LEU2 his3::LacR-GFP:HIS3 TetR-mRFP HHT2S10D:URA3 smc2-8 |
| 10858, 10859, 10860 | a | trp1::TetO:TRP1 lys4::LacO:LEU2 his3::LacR-GFP:HIS3 TetR-mRFP HHT2S10D:URA3 hst2::NatMX ura3 ade1 leu2 |
| 10861, 10862, 10863 | a | trp1::TetO:TRP1lys4::LacO:LEU2 his3::LacR-GFP:HIS3 TetR-mRFP HHT2S10D:URA3 smc2-8 hst2::NatMX |
| 11566, 11567, 11568 | alpha | his3Δ200 leu2Δ0 lys2Δ0 trp1Δ63 ura3Δ0 met15Δ0 can1::MFA1pr-HIS3 hht1-hhf1::NatMX4 hht2-hhf2::[HHTS10AS]-URA3 smc2-8 |
| 11767, 11768, 11769 | a, alpha | his3Δ200 leu2Δ0 lys2Δ0 trp1Δ63 ura3Δ0 met15Δ0 can1::MFA1pr-HIS3 hht1-hhf1::NatMX4 hht2-hhf2::[HHTS-HHFS]-URA3 smc2-8 |
| 12155, 12156, 12157 | a | trp1::TetO:TRP1 lys4::LacO:LEU2 his3::LacR-GFP:HIS3 TetR-mRFP hst2::KanMX |
| 12731, 12732, 12733 | a | his3Δ200 leu2Δ0 lys2Δ0 trp1Δ63 ura3Δ0 met15Δ0 can1::MFA1pr-HIS3 hht1-hhf1::NatMX4 hht2-hhf2::[HHT-S10D-HHFS]-URA3 smc2-8 |
| 13100,13102 | a | trp1::TetO:TRP1 lys4::LacO:LEU2 his3::LacR-GFP:HIS3 TetR-mRFP Hhf1-Hht1::NatMX hht2-hhf2::[HHT-HHFK16R]-URA3 hst2::hphNT1 ura3 ade1 leu2 |

Phosphate Buffered Saline (PBS) containing 0.5 µg/ml DAPI. Cells were prepared on a glass slide and imaged by using the DAPI channel with 300 ms exposure. Transmission images were taken for reference. At least 240 cells were counted for each condition.

## Crosslinking and FACS analysis

Crosslinking and acid extraction of histone H2A was performed and analyzed as previously described (*Wilkins et al., 2014*). In brief, the histone H2A Y58TAG expression vector, under control of its native promoter, was cotransformed into yeast cells with the plasmid pESC-BPARS, which harbors the evolved *E. coli* Tyrosine tRNA$_{CUA}$/amino-acyl tRNA synthetase (Tyr-tRNA$_{CUA}$/AARS) pair, under control of constitutive promoters. Cells were cultured as described below and then subjected to irradiation at 365 nm UV light at a distance of ~5 cm for 7 min on ice (Vilber Lourmat lamp, 2 X 8W, 365 nm tubes, 32 W, 230 V #VL-208.BL). The nuclei of the crosslinked samples were isolated and then the histones further purified by acid extraction. The histone crosslinked products, or H4 K16ac levels, were visualized by western blot chemoluminescence after being decorated with H4 (Abcam, ab7311) or H4 K16ac (Active Motif, 39167) antibodies.

Synchronization of yeast cells was performed in either BY4741 (WT) or *smc2-8* temperature sensitive cells (*Li et al., 2011*). Both cell types were cultured in the same manner except that the *smc2-8* cells were grown at a permissive temperature of 25°C rather than 30°C. Cells were initially inoculated into an overnight culture of standard synthetic complete medium (1.7 g/L Difco Yeast nitrogen base without amino acids, 5 g/L ammonium sulfate, 2% glucose and 2 g/L amino acid dropout mixture) supplemented with p-benzoyl-L-phenylalanine (pBPA) at a final concentration of 1 mM. The cultures were then diluted to an OD$_{600}$ = 0.25 in YPD, also supplemented with pBPA. Cells were allowed to double once at permissive temperatures and then alpha-factor was added to a final concentration of 5 µg/mL (Sigma T6901). Cells were again grown at permissive temperature for 2 hr and the arrest efficiency was monitored by microscopy (arrest stopped when >95% of the cells entered G1 based on morphology). Cells were washed with YPD and then expanded into YPD (without pBPA). At release time zero, nocodazole was added to a final concentration of 1.5 µg/mL and shifted to the restrictive temperature of 37°C. Proper cell arrest and synchronization was monitored by FACS. FACS samples were prepared as previously described (*Haase, 2004*).

## Structured illumination microscopy

Cells were grown in liquid YPD at 30°C and fixed for 15 min in 4% paraformaldehyde (Sigma-Aldrich) and 3.4% sucrose. Fixed cells were spun and the pellet was washed twice with a potassium phosphate buffer (pH 7.5) containing 1.2 M sorbitol. For imaging, cells were spun and resuspended in Vectashield (Vector laboratories) mounting medium for fluorescence. Cells were placed between an unfrosted slide and a number 1.5 high precision coverslip (Marienfeld Superior) sealed with nail polish.

Acquisitions were made with an Applied Precision OMX Blaze (GE Healthcare) equipped with a 60X 1.42 NA Plan Apo oil objective and a sCMOS OMX V4 camera. The oil (Applied Precision) had a refractive index of 1.514. For GFP imaging a 488 nm laser line was used for excitation with a 504–552 nm emission filter. With 10% of the light intensity 300 ms exposures were performed. Stacks span 1.75 µm with a spacing of 125 nm between each focal plane. Image reconstruction was performed with the Softworx software (Applied Precision) with a Wiener filter of 0.002 and a channel specifically measured optical transfer function.

## Acknowledgements

The authors thank K Weis and E Dultz for critical discussion of the data and the Barral, Kroschewski and Matos labs for helpful discussions and comments, M Bayer for help with FRET, F Caudron and the Scientific Center for Optical and Electron Microscopy (ScopeM) of the Swiss Federal Institute of Technology (ETHZ) and especially Tobias Schwarz for microscopy support. This work was supported by the ETH Zurich and an advanced ERC grant to YB.

# Additional information

## Funding

| Funder | Author |
|--------|--------|
| ETH Zurich | Tom Kruitwagen<br>Annina Denoth-Lippuner<br>Yves Barral |
| European Research Council | Tom Kruitwagen<br>Annina Denoth-Lippuner<br>Yves Barral |

The funders had no role in study design, data collection and interpretation, or the decision to submit the work for publication.

## Author contributions

TK, Acquisition of data, Analysis and interpretation of data, Drafting or revising the article; ADL, HN, Analysis and interpretation of data, Drafting or revising the article; BJW, Acquisition of data, Analysis and interpretation of data; YB, Conception and design, Analysis and interpretation of data, Drafting or revising the article

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
