## [Decision Letter]

Thank you for submitting your work entitled "Axial contraction and short-range compaction of chromatin synergistically promote mitotic chromosome condensation" for peer review at *eLife*. Your submission has been favorably evaluated by Jim Kadonaga (Senior editor) and three reviewers, one of whom is a member of our Board of Reviewing Editors.

The reviewers have discussed the reviews with one another and the Reviewing editor has drafted this decision to help you prepare a revised submission.

Summary:

Two of the three referees were enthusiastic about the imaging based approach to study chromosome compaction and axial contraction, while referee 3 was critical of some aspects of your experimental design. Referees 1 and 3 also raised questions about the statistical significance of some of your experiments, which are in points 1 and 4 of referee 3's detailed comments. Referees 2 and 3 have articulated important experimental controls and 1-2 additional experiments that will strengthen your study. The referee comments are attached verbatim.

Essential revisions:

In looking through the comments, it appears that there are only 4-5 points for which experiments are required (those in the comments of referee 2 and 3). These are all necessary to be performed to establish and strengthen your model fully. The super-resolution study that referee 1 suggests is not essential, but if you have STORM or PALM data, in particular, available you may add it to the revision. It is clear that SIM may not be a great improvement over your Deltavision acquired data with appropriate image analysis.

*Reviewer #1:*

This manuscript by Barral and colleagues addresses a fundamental question in chromosome biology, i.e. of the mechanism of chromosome condensation during mitosis. The authors use a clever microscopy based assay to investigate two different processes that contribute to condensation, i.e. of short-range compaction and axial contraction. Compaction, they show, clearly involves histone-histone interactions and is not perturbed in condensin mutants. By contract, axial contraction involves condensin (this was known previously), but they find a new role Hst2 deacetylase in modulating condensin function. They propose a pathway in which two synergistic arms contribute to chromosome condensation during mitosis.

While I find the approach very unique and the conclusions interesting, the statistics in many of the figures make me wonder how significant the various measurements are, especially of the intensities and distance measurements.

I support publication, but would like the following major issue addressed.

With tremendous advances in light microscopy, perhaps they can increase the resolution and intensity measurement/counting using either a SIM (increases resolution to 125 nm) or even better STORM/PALM, which further increases resolution.

While this may sound excessive, this sort of approach is becoming very routine and even getting one data set in wild-type cells for fluorescence intensity difference between G1 and anaphase and similarly the cell cycle dependent differences between *LYS1* and *TRP1* loci will go a long way to convince readers of this important mechanism arrived at using a novel strategy. I want to stress that the intention is not to make the authors repeat the entire study using super-resolution, but to at least set the stage so that we can be sure about the conclusions made using microscopy approaches that do not allow imaging in high resolution.

*Reviewer #2:*

Previous studies have shown that phosphorylation of H3S10 recruits the deacetylase Hst2, which deacetylates H4K16, and promotes the interaction between H2A and H4, a key mechanism in chromatin compaction. The authors have now developed an ingenious method to monitor chromosome compaction by monitoring fluorescence proteins localized to the TetO or LacO arrays, which show quenching when chromatins become compact. Using this tool, they show that condensin is dispensable for chromosome compaction, but indispensable for chromosome contraction. On the other hand, the pathway required for chromosome compaction is mostly indispensable for chromosome contraction. Thus, the authors propose that there are two mechanisms of mitotic chromosome condensation in budding yeast; one is the chromatin compaction mediated by nucleosome-nucleosome interaction, and the other is chromosome contraction mediated by condensin. Most experiments are well controlled and the conclusions are interesting. However, there are some aspects of the paper that weaken the overall story and that could be improved.

1) This manuscript has resolved important issues regarding chromosome condensation regulated by aurora B/Ipl1 in budding yeast. However, it is peculiar that the authors neither analyzed nor mention the contribution of Ipl1 to chromosome compaction/contraction. In fact, the authors previously mention that 'ipl1 inactivation did not abolish anaphase condensation in general (Science, 332, 465-468, 2011)'. How do they reconcile the model shown here (Figure 7) with the previous description? This issue should be addressed, probably by performing additional experiments to examine compaction and contraction under ipl1 inactive conditions.

2) This manuscript stresses the importance of H3S10 phosphorylation by aurora B, although the *H3S10A* mutant shows defects only in compaction. In the final model (Figure 7), the authors suggest that aurora B regulates chromosome contraction through condensin. In fact, condensin phosphorylation by aurora B and its requirement for anaphase condensation has been demonstrated in fission yeast and humans (Tada et al. Nature 2011). Also in budding yeast, Lavoie et al. (Gene Dev 2004) reported the importance of Ipl1 for anaphase condensation. These papers would be cited in the context of the authors' own results.

3) The previous report indicated that Hst2 recruitment to nucleosomes is dependent on *H3S10p*, and acts in the deacetylation of H4K16 and chromosome compaction. Importantly, the H3S10A mutation causes defects only in chromosome compaction, but not in contraction (Figure 1 and Figure 5). Curiously, *H3S10D* (and recruited *Hst2*) not only sustains compaction, but also restores contraction defects caused by a condensin mutation. These facts cannot be reconciled with the authors' interpretation (Figure 7). I assume that although *H3S10D* might cause artificial overloading of Hst2, its loading is usually limited in wild type cells; *H3S10p*-independent (or residual) Hst2 may act in the condensin contraction pathway.

4) It is not clear what happens in *H3S10D*. What happens in *H4∆9-16 H3S10D* in Figure 1?

5) Please examine *H4K16R* and *H4K16Rhst2*∆ in Figure 1, Figure 5 and 6. These experiments will clarify the effect of compaction (but not deacetylation) on contraction.

*Reviewer #3:*

The manuscript of Kruitwagen et al. exploits budding yeast to examine the contributions of short-range nucleosome compaction and long-range condensin-dependent chromosome contraction. By assessing changes in compaction and contraction in mutants defective in condensin and histone modification, the authors conclude that compaction and contraction happen at different times in the cell cycle and suggest potential different modes of regulation. The topic of chromosome condensation in mitosis is important, as chromosome structure in mitosis remains one of the major unanswered questions in biology. Much of the work here seems to corroborate a previous paper published by Wilkins et al. in Science, including the role of H3 and H4 modification in nucleosome compaction, and the role of H3 S10 and Hst2 in nucleosome compaction and chromosome contraction. In addition the manuscript has some important technical and experimental issues that need to be addressed.

1) The authors use an unusual assay to assess compaction, the reduction in fluorescence of TetR-mCherry and LacO-GFP bound to operator arrays between G1 and late anaphase cells. Many possible explanations exist for changes in fluorescence, one of which is compaction. Others include changes in chromatin composition or structure other than compaction that would be equally affected by histone modification. For example, the reduction in fluorescent signal in anaphase is not nearly as pronounced at the *LYS4* locus as it is at the *TRP1* locus. No statistical significance is given to the small drop in fluorescence at the *LYS4* locus. This difference between loci opens up the possibility that the reduction in signal at *TRP1* represents histone-dependent modification of a special feature of the pericentromere region and/or the kinetochore that is well documented by the Bloom lab. The key to believing these modifications represent global change in nucleosome compaction comes from the UV cross-linking experiment of H2-H4. But the recovery of this crosslink is very inefficient. This poor recovery may reflect inefficiencies of the method as the authors have suggested previously, or that the crosslinks are constrained to a small region of the genome, like the unusual pericentric structure.

2) The measurement of chromosome contraction using differentially marked loci on chromosome IV is also technically flawed. It is important to remember that yeast undergoes a closed mitosis such that the nuclear envelope constrains the ability of sequences to separate in space. In G1-metaphase the nucleus is a small ball, which prevents the chromosomes from extending. During anaphase, the nucleus elongates with the elongating spindle before returning to a small ball in late anaphase. As chromosomes are dragged during anaphase in all organisms, they lag behind the centromere and are stretched. In yeast with the elongated nucleus this stretching can manifest itself. When the nucleus shrinks the chromosomes become more confined again. This pattern fits the kinetics of spot separation in Figure 3 exactly. Thus it is unclear whether the kinetics of contraction is meaningful, that is the apparent additional contraction in anaphase from that already in anaphase is real. This worry challenges their conclusion that nucleosome compaction precedes contraction. It would also be important to know that whether the small changes in distances between spots reflects a difference in nuclei morphology due differences in the time of the cell cycle. The authors could include a marker for the nuclear envelope.

3) The authors state, "As expected from previous studies (Neurohr et al., 2011; Lavoie, B. D., Hogan, E., and Koshland), the mutations *H3 S10A* and *H4 Δ9-16* did not impair anaphase contraction (Figure 5), establishing that phosphorylation of H3 S10 and H4/H2A interactions are not necessary for axial contraction of chromosomes. In contrast, however, the *hst2∆* mutation did abrogate the proper contraction of the chromosome during mitosis." Yet this same first author is an author on a previous publication (2014) that states: "Both HST2 inactivation and the H3 S10A mutation impaired this process to the same extent, either alone or in combination (Figure 4)." These results appear contradictory.

4) The authors also conclude that yeast does not cleanse TetR or LacO based upon a failure to see an increase in background fluorescence. If a very small percentage of TetR-mCherry is associated with chromosomes, one is very unlikely to see a change in the background fluorescence if half of it is removed in anaphase. Without a measure of the fraction of the pool in the nucleus that has bound TetR-mCherry this experiment is meaningless.

[Editors' note: further revisions were requested prior to acceptance, as described below.]

Thank you for resubmitting your work entitled "Axial contraction and short-range compaction of chromatin synergistically promote mitotic chromosome condensation" for further consideration at *eLife*. Your revised article has been favorably evaluated by Randy Schekman (Senior editor) and three reviewers, one of whom, (Mohan Balasubramanian) is a member of our Board of Reviewing Editors. The manuscript has been improved but there are some remaining textual issues that need to be addressed before acceptance, as outlined below.

As you will see, two of the original three referees raise minor points to do with: 1) deleting some of the kinetic analyses of chromosome contraction (referee 3) and 2) making the model figure clear and also streamlining the manuscript to clarify the role of H3 S10 phosphorylation (referee 2).

These should not take you very long and we will be pleased to accept the further revised version.

Please see the referees' minor comments below.

*Reviewer #2:*

The authors appear to have addressed most of the points I raised in the initial round of review. I still have, however, one major issue.

The authors' interpretation concerning the *H3S10D* phenotype is now under question. I agree that *H3S10D* over-recruits Hst2. Although this mutant restores chromosome contraction in some assays, there is no defect in physiological chromosome contraction in the *H3S10A* mutant in which Hst2 is not recruited by H3 (Figure 5 and Figure 6). Therefore, it is most likely that *H3S10D* causes either 'over compaction', which by itself somewhat shortens the axis length, or artificial enhancement of the condensin-dependent contraction. The latter case is now properly represented in Figure 7 (left panel) by adding a new Hst2 out of *H3S10p*. This pool of Hst2 might be artificially supplied when *H3S10D* is expressed. Therefore, the current results do not provide any evidence to support the notion that 'H3S10 phosphorylation is involved in condensin regulation', at least under physiological conditions. This point is very confusing throughout the manuscript. I strongly recommend that the arrows in (*H3S10p*) Hst2  condensin  contraction in Figure 7 (left panel) be removed, as well as the related content in the text. Instead, I agree that the simple model presented in Figure 7 (right panel) that clarifies the concept of compaction (by H3S10 phosphorylation) and contraction (by condensin), which act synergistically to form a properly condensed chromosome.

*Reviewer #3:*

The authors have significantly strengthened the manuscript, having addressed many of the important issues.

However they missed the point of the criticism about the analyzing the kinetics of chromosome contraction. The authors clearly have shown that they observe differences in contraction between late anaphase and G1 cells that are independent of nuclear morphology. However, they have not shown that the apparent longer distances between their arrays in early and mid-anaphase, which they assume to be less contraction, are not a consequence of changes in nuclear morphology. The chromosome stretching during chromosome movement may reveal the real distance between the arrays along the chromosome axis. When confined to a small round nucleus the distances between the arrays represent the shortest distance between arrays on bent chromosomes. In other words if one could make early and mid-anaphase cells that have small round nuclei like late anaphase and G1 cells, the distances between the arrays would shrink and be the same as late anaphase cells, all of which would be shorter than G1 cells. So contraction is greater in late anaphase than G1 as the authors contend but it is also equally greater in early anaphase and mid anaphase than G1. If so, contraction is happening at the onset of early anaphase concurrent with compaction and not later than it. I can't think of a way for the authors to address this nuclear morphology issue that really dramatically hinders any conclusion from this experimental strategy about contraction in early and mid-anaphase. However this kinetic analysis is really a side embellishment of the major points of the manuscript and as such deleting this section will not detract from the manuscript.

---

## [Author Response]

Reviewer #1:

*This manuscript by Barral and colleagues addresses a fundamental question in chromosome biology, i.e. of the mechanism of chromosome condensation during mitosis. The authors use a clever microscopy based assay to investigate two different processes that contribute to condensation, i.e. of short-range compaction and axial contraction. Compaction, they show clearly involves histone-histone interactions and is not perturbed in condensin mutants. By contract, axial contraction involves condensin (this was known previously), but they find a new role Hst2 deacetylase in modulating condensin function. They propose a pathway in which two synergistic arms contribute to chromosome condensation during mitosis. While I find the approach very unique and the conclusions interesting, the statistics in many of the figures make me wonder how significant the various measurements are, especially of the intensities and distance measurements.*

We agree with the reviewer that our data on chromatin compaction and contraction is of a highly variable nature. To investigate this variability, we followed individual dividing cells and measured the fluorescence intensity at the LacO/LacI-GFP locus (which was more resistant to photobleaching than the TetO/TetR-mCherry focus). The result of this experiment is now shown in Figure 3 of our revised manuscript. We discuss these data as follows in the Results section:

“Next, we investigated the dynamics of chromatin compaction during the cell cycle. To this end, we visualized both the TetO/TetR-mCherry (at *TRP1*) and LacO/LacI-GFP (at *LYS4*) loci simultaneously. […] In either case, this intrinsic cell-to-cell variability precludes drawing conclusions at the single cell level and emphasizes the fact that the quenching assay introduced here is statistical in nature.”

*I support publication, but would like the following major issue addressed.*

*With tremendous advances in light microscopy, perhaps they can increase the resolution and intensity measurement/counting using either a SIM (increases resolution to 125 nm) or even better STORM/PALM, which further increases resolution. While this may sound excessive, this sort of approach is becoming very routine and even getting one data set in wild-type cells for fluorescence intensity difference between G1 and anaphase and similarly the cell cycle dependent differences between LYS1 and TRP1 loci will go a long way to convince readers of this important mechanism arrived at using a novel strategy. I want to stress that the intention is not to make the authors repeat the entire study using super-resolution, but to at least set the stage so that we can be sure about the conclusions made using microscopy approaches that do not allow imaging in high resolution.*

Indeed, super resolution microscopy could be a way to confirm and gain more insight into the specific structures that underlie, in particular, chromatin compaction. The reviewer mentions three “standard” methods that can be used. For PALM, specific antibodies against, in our case, TetR- mCherry or LacI-GFP could be applied. Indeed, we have tried this in the past (unpublished) to visualize equivalent TetO-containing chromatin using PALM, but we encountered technical problems that would require significant optimization. Thus, this approach currently lies beyond the scope of this manuscript. STORM could be a great, equivalent, method, given that we construct the required photo-switchable or -activatable TetR fusion proteins. However, so far we have not been successful with using STORM in yeast cells either. Given the nature of this method, however, we predict that we would not see an increase in fluorophore-quenching in anaphase, since homo-fluorophores are not excited at the same time anymore. It can only be used as a method to measure the size of the LacO and TetO loci and test whether they indeed compact during anaphase. This might already be tested using SIM, which is currently available to us.

Thus, we fixed cells and used an in-house Applied Precision OMX Blaze to visualize LacO/LacI- GFP foci. Indeed, we observed that anaphase foci are dimmer and smaller than their G1 counterparts. However, upon careful analysis of the intensity profile, we observed that these still reflect a simple Gaussian distribution. Thus, the resolution provided by SIM is probably not sufficient to measure the size of the locus and compare between G1 and anaphase. Furthermore, the highly processive nature of the image reconstruction might result in artifacts when combined with quenching. Based on this preliminary experiment we conclude that the changes in fluorescence loci caused by chromatin compaction cannot be resolved by SIM, but rather require optimization of STORM/PALM. We are sorry that we cannot provide the analysis wished by the reviewer at this stage.

Reviewer #2:

*Previous studies have shown that phosphorylation of H3S10 recruits the deacetylase Hst2, which deacetylates H4K16, and promotes the interaction between H2A and H4, a key mechanism in chromatin compaction. The authors have now developed an ingenious method to monitor chromosome compaction by monitoring fluorescence proteins localized to the TetO or LacO arrays, which show quenching when chromatins become compact. Using this tool, they show that condensin is dispensable for chromosome compaction, but indispensable for chromosome contraction. On the other hand, the pathway required for chromosome compaction is mostly indispensable for chromosome contraction. Thus, the authors propose that there are two mechanisms of mitotic chromosome condensation in budding yeast; one is the chromatin compaction mediated by nucleosome-nucleosome interaction, and the other is chromosome contraction mediated by condensin. Most experiments are well controlled and the conclusions are interesting. However, there are some aspects of the paper that weaken the overall story and that could be improved. 1) This manuscript has resolved important issues regarding chromosome condensation regulated by aurora B/Ipl1 in budding yeast. However, it is peculiar that the authors neither analyzed nor mention the contribution of Ipl1 to chromosome compaction/contraction. In fact, the authors previously mention that 'ipl1 inactivation did not abolish anaphase condensation in general (Science, 332, 465-468, 2011)'. How do they reconcile the model shown here (Figure 7) with the previous description? This issue should be addressed, probably by performing additional experiments to examine compaction and contraction under ipl1 inactive conditions.*

In our previous work (Neurohr et al., 2011), Ipl1 was inactivated shortly before anaphase onset by switching cells containing the *ipl1-321* temperature sensitive allele to the restrictive temperature accordingly. With this protocol, we observed a minor and actually insignificant defect in wild type chromosome contraction (whereas the contraction of the artificially long chromosome was significantly decreased). We now used the same strain, but inactivated Ipl1 throughout mitosis, i.e. shifted the cells to the restrictive temperature while they were still in G1. Strikingly, this resulted in a significant failure in both chromosome compaction and contraction (Figure 1 and Figure 5), as reported before (Lavoie et al., 2004).

These data indicate that for normal sized, wild type chromosomes, Ipl1 promotes condensation (compaction and condensation) prior to anaphase onset, probably during prophase and/or metaphase. This is in fact suggests that there are intermediate steps between Ipl1 function in early mitosis and the actual process of full condensation, which is most obvious during anaphase. Therefore, our data suggest that Ipl1 functions essentially in priming chromosomes for condensation. Subsequently, a second step downstream of Ipl1 is required to implement actual chromosome contraction and this second step can be temporally separated from the initial step (i.e. pro/metaphase and early anaphase). For example, the phosphorylation of condensin by Ipl1 may contribute to its loading or prime it for activation, whereas its phosphorylation by Cdk1 might delay its full activation for its role in chromosome contraction (Robellet et al., 2015). In such a scenario, full contraction would rely on Ipl1 activity in metaphase and activation of FEAR, activation of Cdc14 and condensin dephosphorylation on its Cdk1 sites during anaphase. In this scenario, the effects of mutating the Cdk1 sites in condensin might reflect consequences of premature activation rather than lack of activation.

*2) This manuscript stresses the importance of H3S10 phosphorylation by aurora B, although the* H3S10A *mutant shows defects only in compaction. In the final model (Figure 7), the authors suggest that aurora B regulates chromosome contraction through condensin. In fact, condensin phosphorylation by aurora B and its requirement for anaphase condensation has been demonstrated in fission yeast and humans (Tada et al. Nature 2011). Also in budding yeast, Lavoie et al. (Gene Dev 2004) reported the importance of Ipl1 for anaphase condensation. These papers would be cited in the context of the authors' own results.*

The reviewer is right. These citations are now incorporated in the Introduction and Discussion.

*3) The previous report indicated that Hst2 recruitment to nucleosomes is dependent on H3S10p, and acts in the deacetylation of H4K16 and chromosome compaction. Importantly, the H3S10A mutation causes defects only in chromosome compaction, but not in contraction (Figure 1 and Figure 5). Curiously, H3S10D (and recruited Hst2) not only sustains compaction, but also restores contraction defects caused by a condensin mutation. These facts cannot be reconciled with the authors' interpretation (Figure 7). I assume that although H3S10D might cause artificial overloading of Hst2, its loading is usually limited in wild type cells; H3S10p-independent (or residual) Hst2 may act in the condensin contraction pathway.*

We fully agree with the reviewer’s interpretation of the data, and it is in fact what we aimed to suggest in Figure 7. While H3 S10 phosphorylation has no role in chromosome contraction in the majority of WT mitoses, *H3 S10D* mutant cells indeed show constitutively condensed chromosomes and this depends on Hst2 activity. *H3 S10D* in fact suppresses the effect of the *smc2-8* mutation in condensin, through a yet unknown mechanism, which depends on Hst2. We indeed suggest that increased activation of Hst2 in the *H3 S10D* mutant cells hyper-activates the (crippled) condensin or acts in a still unidentified contraction pathway.

As the reviewer proposes, our data indicate that only a fraction of H3 S10 is phosphorylated in most WT mitoses, and some Hst2 must be indeed recruited to chromatin independently of H3 S10 phosphorylation to promote chromosome contraction. This was certainly not evident in our previous model figure, which we have amended to clarify this point.

Lastly, in the presence of an artificially long chromosome, such as in Neurohr et al. (2011), we propose that increased phosphorylation of H3 S10 by Ipl1 on the midzone boosts up the activation of Hst2 on that chromosome beyond regular levels, promoting its further contraction to fit it into the spindle. We have clarified the Results and Discussion sections regarding these issues.

*4) It is not clear what happens in H3S10D. What happens in H4∆9-16 H3S10D in Figure 1?*

See below.

*5) Please examine H4K16R and H4K16R hst2∆ in Figure 1, Figure 5 and 6. These experiments will clarify the effect of compaction (but not deacetylation) on contraction.*

We thank the reviewer for raising these points, which together raise the issue of whether acetylated K16 of H4 is the main target of Hst2 upon H3 S10 phosphorylation, in which case *H4∆9-16* should bypass the effect of *H3 S10D* and *H4 K16R* should bypass the compaction phenotype of the *hst2*∆ mutation but not its contraction phenotype. Due to the organization of the histone loci in budding yeast, constructing the *H4∆9-16 H3 S10D* is challenging and we have therefore addressed the issue by focusing on characterizing the *H4 K16R hst2*∆ double mutant.

From our original data, it was already clear that the *H4 K16* acetylation alone does not affect chromosome contraction, which still takes place normally in the *H4 ∆9-16* mutant cells.

Supporting this idea further, the *H4 K16R hst2*∆ double mutant does not contract its chromosome IV (Figure 5), indicating that deacetylation of K16 on H4 is not sufficient to restore contraction in the *hst2*∆ mutant cells. Thus, we conclude that: 1) H4 deacetylation has no evident effects on chromosome contraction and 2) that Hst2, when deleted, does not have its defects on chromosome contraction through the chromatin compaction pathway, but rather by acting on another factor in the contraction pathway.

The picture is more complex for what concerns chromosome compaction. Indeed, unlike in the *H4 K16R* single mutant fluorescence intensity at the TetO/TetR-mCherry focus remained continuously high over the cell cycle in the *H4 K16R hst2*∆ double mutant cells, analogous to what is observed in the *hst2*∆ single mutant cells (Figure 1). This implies that Hst2 promotes chromatin compaction not solely through deacetylation of K16 on H4 but also through additional downstream targets. The nature of these targets is not known at this stage but could include K8 and K12 on H4. Addressing the role of these residues and identifying possible additional targets of Hst2 will be the focus of further studies.

Reviewer #3:

*The manuscript of Kruitwagen et al. exploits budding yeast to examine the contributions of short-range nucleosome compaction and long-range condensin-dependent chromosome contraction. By assessing changes in compaction and contraction in mutants defective in condensin and histone modification, the authors conclude that compaction and contraction happen at different times in the cell cycle and suggest potential different modes of regulation. The topic of chromosome condensation in mitosis is important, as chromosome structure in mitosis remains one of the major unanswered questions in biology. Much of the work here seems to corroborate a previous paper published by Wilkins et al. in Science, including the role of H3 and H4 modification in nucleosome compaction, and the role of H3 S10 and Hst2 in nucleosome compaction and chromosome contraction. In addition the manuscript has some important technical and experimental issues that need to be addressed. 1) The authors use an unusual assay to assess compaction, the reduction in fluorescence of TetR-mCherry and LacO-GFP bound to operator arrays between G1 and late anaphase cells. Many possible explanations exist for changes in fluorescence, one of which is compaction. Others include changes in chromatin composition or structure other than compaction that would be equally affected by histone modification. For example, the reduction in fluorescent signal in anaphase is not nearly as pronounced at the LYS4 locus as it is at the TRP1 locus. No statistical significance is given to the small drop in fluorescence at the LYS4 locus. This difference between loci opens up the possibility that the reduction in signal at TRP1 represents histone-dependent modification of a special feature of the pericentromere region and/or the kinetochore that is well documented by the Bloom lab. The key to believing these modifications represent global change in nucleosome compaction comes from the UV cross-linking experiment of H2-H4. But the recovery of this crosslink is very inefficient. This poor recovery may reflect inefficiencies of the method as the authors have suggested previously, or that the crosslinks are constrained to a small region of the genome, like the unusual pericentric structure.*

We thank the reviewer very much for raising this important question. To address whether the *TRP1* locus reflects a specialized chromatin region that is not representative for the whole chromosome, we analyzed the fluorescence intensity of LacO/LacI-GFP at the *LYS4* locus in the middle of the chromosome arm more carefully. Though it might compact slightly less than *TRP1,* we still observe a significant difference in fluorescence intensity between G1 and anaphase cells at the *LYS4* locus. Furthermore, analysis of the LacO/LacI-GFP brightness in the *H3 S10A, H3 S10D, hst2*∆ and *H3 S10D hst2*∆ single and double mutants confirms that H2A/H4 interactions are also required to compact this *CEN4* distal locus. These data indicate that chromatin compaction is taking place over the whole chromosome. They are now added to Figure 1 where they extend the TetO/TetR-mCherry data from Figure 1. However, since the anaphase specific quenching is not as strong as it is at the *CEN4* proximal *TRP1* locus, we do not exclude that the specialized chromatin domain around centromeres reaches a tighter compaction state during mitosis.

*2) The measurement of chromosome contraction using differentially marked loci on chromosome IV is also technically flawed. It is important to remember that yeast undergoes a closed mitosis such that the nuclear envelope constrains the ability of sequences to separate in space. In G1-metaphase the nucleus is a small ball, which prevents the chromosomes from extending. During anaphase, the nucleus elongates with the elongating spindle before returning to a small ball in late anaphase. As chromosomes are dragged during anaphase in all organisms, they lag behind the centromere and are stretched. In yeast with the elongated nucleus this stretching can manifest itself. When the nucleus shrinks the chromosomes become more confined again. This pattern fits the kinetics of spot separation in Figure 3 exactly. Thus it is unclear whether the kinetics of contraction is meaningful, that is the apparent additional contraction in anaphase from that already in anaphase is real. This worry challenges their conclusion that nucleosome compaction precedes contraction. It would also be important to know that whether the small changes in distances between spots reflects a difference in nuclei morphology due differences in the time of the cell cycle. The authors could include a marker for the nuclear envelope.*

To address the possible role of nuclear morphology in chromosome condensation, we measured the nuclear diameter in G1 and late anaphase nuclei in WT and *hst2*∆ conditions. We failed to observe any significant difference between G1 and anaphase in either strain. If anything, late anaphase nuclei tended to be a bit bigger than their G1 counterparts, rather opposing than facilitating chromosome condensation. Thus, we conclude that nuclear morphology during mitosis is not a major determinant of the distance measured between *TRP1* and *LYS4* in *WT* cells. Rather, these data support the conclusion that this distance reflects changes in axial chromosome contraction, as suggested by the numerous studies that have relied on this assay by now.

3) The authors state, "As expected from previous studies (Neurohr et al., 2011; Lavoie, B. D., Hogan, E., and Koshland), the mutations H3 S10A and H4 Δ9-16 did not impair anaphase contraction (Figure 5), establishing that phosphorylation of H3 S10 and H4/H2A interactions are not necessary for axial contraction of chromosomes. In contrast, however, the hst2∆ mutation did abrogate the proper contraction of the chromosome during mitosis." Yet this same first author is an author on a previous publication (2014) that states: "Both HST2 inactivation and the H3 S10A mutation impaired this process to the same extent, either alone or in combination (Figure 4)." These results appear contradictory.

It is important to note the differences between the wild type situation investigated in the presentstudy, where we have looked exclusively at regular chromosomes, and the study to which the reviewer is referring in which we characterized specifically the effect of Hst2 and H3S10 on the contraction of an artificially long chromosome, such as those constructed by Neurohr et al. (2011). Indeed, the 2014 study focused on the phenomenon of adaptive hypercondensation.

Therefore, the differences pointed out by the reviewer between the data obtained in the present study and those reported in the 2014 paper reflect how much hypercondensation is a phenomenon distinct from regular chromosome organization. Regular chromosomes do not depend on H3 S10Ph to contract, whereas the artificially long chromosome does. We showed that adaptive hypercondensation depends on the anaphase activity of the Ipl1 kinase, its midzone localization, H3 S10Ph and on Hst2 function (Neurohr et al., 2011; Wilkins et al., 2014), whereas regular condensation relies on metaphase activity of Ipl1, and Hst2 function. We propose that, in the case of a longer chromosome, more H3 S10 is phosphorylated by midzone Ipl1 in anaphase, which leads to an increased activation of Hst2. As explained in our response to reviewer 2 (point 3), we propose that increase Hst2 activity on chromatin leads to activation of its substrates in the contraction pathway, such as perhaps condensin. In any case, the long chromosome depends more heavily on H3 S10 phosphorylation, compared to WT chromosomes. To clarify these points, we added some explanations to the Discussion section regarding the difference between regular chromosome condensation and adaptive hypercondensation.

*4) The authors also conclude that yeast does not cleanse TetR or LacO based upon a failure to see an increase in background fluorescence. If a very small percentage of TetR-mCherry is associated with chromosomes, one is very unlikely to see a change in the background fluorescence if half of it is removed in anaphase. Without a measure of the fraction of the pool in the nucleus that has bound TetR-mCherry this experiment is meaningless.*

In order to address this issue, we determined that the fraction of bound TetR-mCherry in G1 cells is 0.19. Given this number, it would indeed be difficult to measure any significant change in background fluorescence in the case of cleansing with our current microscopy setup. We therefore thank the reviewer for this useful insight and have removed these data from Figure 2 and changed the text accordingly.

[Editors' note: further revisions were requested prior to acceptance, as described below.]

*As you will see, two of the original three referees raise minor points to do with: 1) deleting some of the kinetic analyses of chromosome contraction (referee 3) and 2) making the model figure clear and also streamlining the manuscript to clarify the role of H3 S10 phosphorylation (referee 2).*

*These should not take you very long and we will be pleased to accept the further revised version. Please see the referees' minor comments below.*

Reviewer #2:

*The authors appear to have addressed most of the points I raised in the initial round of review. I still have, however, one major issue.*

*The authors' interpretation concerning the H3S10D phenotype is now under question. I agree that H3S10D over-recruits Hst2. Although this mutant restores chromosome contraction in some assays, there is no defect in physiological chromosome contraction in the H3S10A mutant in which Hst2 is not recruited by H3 (Figure 5 and Figure 6). Therefore, it is most likely that H3S10D causes either 'over compaction', which by itself somewhat shortens the axis length, or artificial enhancement of the condensin-dependent contraction. The latter case is now properly represented in Figure 7 (left panel) by adding a new Hst2 out of H3S10p. This pool of Hst2 might be artificially supplied when H3S10D is expressed. Therefore, the current results do not provide any evidence to support the notion that 'H3S10 phosphorylation is involved in condensin regulation', at least under physiological conditions. This point is very confusing throughout the manuscript. I strongly recommend that the arrows in (H3S10p) Hst2  condensin  contraction in Figure 7 (left panel) be removed, as well as the related content in the text. Instead, I agree that the simple model presented in Figure 7 (right panel) that clarifies the concept of compaction (by H3S10 phosphorylation) and contraction (by condensin), which act synergistically to form a properly condensed chromosome.*

The reviewer is right that there are two ways to explain the constitutive contraction phenotype of *H3 S10D* mutant cells and both depend on hyper-recruitment of Hst2. On the one hand, over-compaction by maximum nucleosome-nucleosome interaction could start to show its effects in contraction, independently of the contraction pathway. On the other hand, Hst2 might activate components of the chromosome contraction pathway (such as condensin). Our current and past experimental data speak against over-compaction since (1) TetO and LacO foci are not dimmer in *H3 S10D* strains as compared to WT, (2) there is not more crosslinking between H4 and H2A in an *H3 S10D* strain as compared to WT (Wilkins et al., 2014) and (3) H3 S10 phosphorylation, through Hst2, was shown to be required for adaptive hyper-condensation (Neurohr et al., 2011 and Wilkins et al., 2014). Since our data is supportive of Hst2 and condensin being in the same genetic pathway, we favor the idea that Hst2 might promote condensin activation through a yet unknown mechanism. However, it is crucial to realize, as the reviewer does, that the over-recruitment of Hst2 observed in *H3 S10D* strains is not relevant for the vast majority of yeast mitoses. Rather, this scenario becomes relevant in the case of smaller yeast nuclei and/or larger yeast chromosomes that require an additional burst of chromosome condensation. We included a new section to the Discussion to address these points.

Reviewer #3:

*The authors have significantly strengthened the manuscript, having addressed many of the important issues. However they missed the point of the criticism about the analyzing the kinetics of chromosome contraction. The authors clearly have shown that they observe differences in contraction between late anaphase and G1 cells that are independent of nuclear morphology. However, they have not shown that the apparent longer distances between their arrays in early and mid-anaphase, which they assume to be less contraction, are not a consequence of changes in nuclear morphology. The chromosome stretching during chromosome movement may reveal the real distance between the arrays along the chromosome axis. When confined to a small round nucleus the distances between the arrays represent the shortest distance between arrays on bent chromosomes. In other words if one could make early and mid-anaphase cells that have small round nuclei like late anaphase and G1 cells, the distances between the arrays would shrink and be the same as late anaphase cells, all of which would be shorter than G1 cells. So contraction is greater in late anaphase than G1 as the authors contend but it is also equally greater in early anaphase and mid anaphase than G1. If so, contraction is happening at the onset of early anaphase concurrent with compaction and not later than it. I can't think of a way for the authors to address this nuclear morphology issue that really dramatically hinders any conclusion from this experimental strategy about contraction in early and mid-anaphase. However this kinetic analysis is really a side embellishment of the major points of the manuscript and as such deleting this section will not detract from the manuscript.*

We apologize for not recognizing this issue in the first set of reviewers’ comments. The reviewer suggests that especially in early anaphase nuclear morphology might play a bigger role in chromosome condensation, since the chromosomes are less constrained in their movement due to a bigger nuclear volume, naturally causing the TetO-LacO distance to be high, though contraction is already present. Although we definitely consider nuclear morphology to play a role in chromosome extension during the early stages of anaphase, we do not think that this is the only factor contributing to it. Specifically, it has been shown that residual arm cohesion in the early stages of anaphase can cause chromosomes to stretch out during exactly this stage (Harrison et al., 2009 and Tanaka et al., 2010). We believe that this phenomenon is causing the very high TetO-LacO distance in early anaphase cells and would like to propose that initiating contraction too early (i.e. in early in stead of late anaphase) might lead to defects in chromosome segregation. However, to properly address this issue the activity of the compaction and contraction machinery in these cell cycle stages would have to be determined with an alternative method. We have discussed these models and their implications in the Discussion and thank the reviewer for these constructive comments.